# Bounds on Over-Parameterization for Guaranteed Existence of Descent Paths in Shallow ReLU Networks

**Arsalan Sharifnassab**[*]
Department of Electrical Engineering
Sharif University of Technology
Tehran, Iran
`a.sharifnassab@gmail.com`

**Saber Salehkaleybar**[†]
Department of Electrical Engineering
Sharif University of Technology
Tehran, Iran
`saleh@sharif.edu`

**S. Jamaloddin Golestani**
Department of Electrical Engineering
Sharif University of Technology
Tehran, Iran
`golestani@sharif.edu`

## Abstract

We study the landscape of squared loss in neural networks with one-hidden layer and ReLU activation functions. Let $m$ and $d$ be the widths of hidden and input layers, respectively. We show that there exist poor local minima with positive curvature for some training sets of size $n \geq m + 2d - 2$. By positive curvature of a local minimum, we mean that within a small neighborhood the loss function is strictly increasing in all directions. Consequently, for such training sets, there are initialization of weights from which there is no descent path to global optima. It is known that for $n \leq m$, there always exist descent paths to global optima from all initial weights. In this perspective, our results provide a somewhat sharp characterization of the over-parameterization required for "existence of descent paths" in the loss landscape.

## 1 Introduction

We consider shallow neural networks of the form shown in Fig. 1. The network comprises a hidden layer and an input layer of widths $m$ and $d$, respectively; and is to be trained over a training set of size $n$. Our results concern the slightly over-parameterized regime where $n \approx m$. We study the existence of poor local minima that have positive curvature in the empirical squared loss landscape.

It is well-known that poor local minima exist in the loss landscape of shallow networks of arbitrary width. In fact, in a shallow network with ReLU activation functions, it is easy to construct training sets whose empirical loss landscape has high plateaus.[1] It is however not fully understood that under what conditions poor local minima may have positive curvature. This paper presents results that improve this understanding.

Non-existence of spurious local minima is closely connected to the so called *descent path property*: a loss landscape is said to have the descent path property if starting from any initial point there is a path of descent loss to a global minimum. From optimization perspective, the descent path property favors descent optimization algorithms like the pure gradient descent (GD) method. For SGD as well, non-existence of poor local minima is known to be a favorable property for guaranteed convergence (Ge et al., 2015; Jin et al., 2017; Lee et al., 2016). The descent path property is shown to be satisfied in over-parameterized shallow networks with sufficiently large widths (Venturi et al., 2018). The

---

[*]Webpage: `http://ee.sharif.ir/~sharifnassab/`
[†]Webpage: `http://sina.sharif.ir/~saleh/`
[1]e.g., for a training set and a set of weights at which all neurons are inactive.

results we present in this work, tighten the existing bounds on the over-parameterization required to guarantee this property.

## 1.1 BACKGROUND

Over the past few years, deep neural networks have achieved tremendous performance in various artificial intelligence applications such as computer vision, reinforcement learning, and natural language processing, etc. Despite their remarkable success in practice, theoretical aspects of this success remain a mystery. It has long been an open problem why simple local search algorithms for training deep neural networks, like stochastic gradient descent (SGD), typically converge to local minima with low training error despite the highly non-convex behavior of empirical loss. It has been observed, e.g., in (Choromanska et al., 2015), that these methods may get stuck in poor local minima (i.e., local minima with empirical loss much larger than the global optimum) for small networks, while the problem fades away as the number of parameters grows larger. Such observations are often explained by studying the loss landscape in over-parameterized regime where the number of parameters in the network exceeds the training sample size.

Recently, several attempts have been made to characterize properties of squared loss landscape by conditioning on the layers' dimensions and sample size. Soudry and Hoffer (2017) showed that weights of a neural network can be adjusted such that the empirical loss is zero almost surely if $m > 4\lceil n/(2d-2)\rceil \approx (2n)/d$. This result is consistent with experimental observations that neural networks can fit training data if the number of parameters (here approximately $2n$) is greater than the sample size. They also proved for normally distributed input that as $n$ goes to infinity, the ratio between the volume of poor flat local minima regions to the volume of flat global minima fades exponentially if $d = \tilde{\Omega}(\sqrt{n})$ and $m = \tilde{\Omega}(n/d)$. Safran and Shamir (2016) showed that if the number of neurons in the hidden layer is $\Omega(n^{\text{rank}(X)})$ (where $X$ is the matrix containing all input), then with high probability, random initialization of weights will put them in a region of parameter space at which the loss surface has a basin-like structure, i.e., every local minimum in that region is global. In another work (Safran and Shamir, 2017), the same authors provide a computer-assisted proof to show that spurious local minima are common in the expected loss landscape of shallow under-parameterized (small-width) networks. Xie et al. (2016) showed that if the input data is drawn uniformly at random from the unit sphere, and if $m = \tilde{\Omega}(n^{\beta})$ and $d = \tilde{\Omega}(n^{\beta})$ with $\beta \in (0,1)$ being the decay exponent of the smallest eigenvalue of a kernel matrix, then every critical point is a global minimum. Li et al. (2018) proved that for any continuous activation function and under the assumption that data samples are distinct, there exist no poor local minima with positive curvature if $m \geq n$ . In the same spirit, Venturi et al. (2018) showed that for any continuous activation function, there is always a descent path to an optimal solution with zero loss in the empirical loss landscape if $m \geq n$.

Several works have proposed similar results in other settings and under different assumptions. Soudry and Carmon (2016) showed that in a network of leaky ReLU activation functions with randomized perturbation of slopes, all differentiable local minima are global minima if $m \geq n/d$. Kawaguchi (2016) proved that in shallow networks with linear activation functions, every local minimum is a global minimum and all the saddle points are strict in the sense that they have a direction of strictly negative curvature. Soltanolkotabi et al. (2019) showed that the same result carries over to quadratic activation functions under the assumption that the last layer comprises at east $d$ positive and $d$ negative weights. Du and Lee (2018) established similar results for quadratic activation functions, assuming $m \geq \sqrt{2n}$. For deep neural networks with linear activation functions, Freeman and Bruna (2016) showed that all local minima are global minima if there is a hidden layer whose number of neurons exceeds the minimum of the widths of input and output layers. For deep neural networks with analytical activation functions, Nguyen and Hein (2017) proved a similar property under the assumptions that the number of neurons in some hidden layer is greater than sample size and the network has a pyramidal structure.

Such studies on the properties of loss landscape do not only provide insights into the complication of training, but are also beneficial for proving performance guarantees for some local search algorithms. For the class of loss functions whose landscape satisfy the properties of: a) all local minima are global, b) all saddle points are strict, it has been shown in several works (Ge et al., 2015; Jin et al., 2017; Lee et al., 2016) that perturbed gradient descent converges to global optima in polynomial time. Another direction of research concerns the convergence of specific optimization algorithm such as

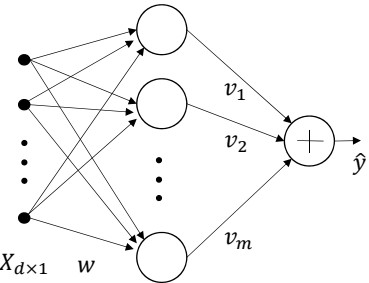

Figure 1: Architecture of the shallow network considered in this paper. The network has a single hidden layer of $m$ neurons with ReLU activation functions, and a neuron with linear activation function in its output layer.

pure gradient descent and SGD without assuming such properties for the loss landscape (Du et al., 2018; Allen-Zhu et al., 2018; Du et al., 2018).

## 1.2 OUR CONTRIBUTIONS

We study the amount of over-parameterization required for guaranteed existence of descent paths to zero loss in the empirical loss landscape. Previous works suggest that zero loss is always possible for $m \geq 2n/d$ (Soudry and Hoffer, 2017). On the other hand, the best existing bound for guaranteed existence of descent paths to this zero loss requires $m \geq n$ neurons in the hidden layer (Venturi et al., 2018). Prior to the present work, it was not known whether the "descent path property" holds for $m < n$. Even for $m \in (2n/d, n)$, where zero empirical risk is known to be achievable (Soudry and Hoffer, 2017), the existence of descent paths was in question. In this work, we tighten this gap and prove that there are training sets, under which in any network of width $m \leq n - 2d + 2$, there exist initial weights that have no descent path to global minima. We do this by showing that the loss landscape, in this regime, admits poor local minima with positive curvature. We also provide evidences and make conjectures that these results carry over to networks of width $m = n - 4$, which, if true, provides a sharp characterization of the over-parameterization required for guaranteed existence of descent paths. We also wish to point that unlike most previous works, we do not restrict to differentiable local minima; for a simple argument shows that local minima with positive curvature cannot be differentiable if $m > n/d$ (cf. Appendix A).

## 1.3 OUTLINE

We continue by discussing details of the system model and introducing our key definitions in Section 2. We then present, in Section 3, the main results of the paper. Proof of the main results are then given in Section 4. We finally discuss implications and possible extensions of our results in Section 5 along with a number of open problems and directions for future research.

## 2 PRELIMINARIES

### 2.1 MODEL

We consider shallow networks of the form shown in Fig. 1. The network takes $d$-dimensional inputs denoted by $X$. There is a single hidden layer comprising $m$ neurons with ReLU activation function. For simplicity of our proofs, we only consider even values of $m$. We denote the input weights of $r$-th neuron by a $d$-dimensional vector $w_r$, for $r = 1, \ldots, m$. We then let $w \in \mathbb{R}^{md}$ be the vector representation of all weights in the first layer.

The output layer has a single neuron, whose activation function is linear with an $m$-dimensional weight vector denoted by $v$. The network outputs a scalar $\hat{y}(w, v) = \sum_{r=1}^{m} v_r w_r^T X \mathbf{1}(w_r^T X \geq 0)$. We fix a training set $(X_1, y_1), \ldots, (X_n, y_n)$ of size $n$, and consider the landscape of empirical squared loss function:

$$F(w, v) \triangleq \sum_{i=1}^{n} \big(\hat{y}_i(w, v) - y_i\big)^2. \tag{1}$$

## 2.2 PROPERTIES OF THE LANDSCAPE

We first provide a formal definition for the descent path property, which is a necessary condition for guaranteed performance of descent optimization algorithms.

**Definition 1** (Descent path property). *Consider a continuous function $f : \mathbb{R}^d \to \mathbb{R}$ and let $f^* = \inf_{x \in \mathbb{R}^d} f(x)$ be its infimum. We say that $f$ has the descent path property if for any $x \in \mathbb{R}^d$, there exists a continuous curve with $\gamma : [0, 1] \to \mathbb{R}^d$ such that $\gamma(0) = x$, $f(\gamma(1)) = f^*$, and $f(\gamma(t))$ is a non-increasing function of $t$.*

The descent path property is a necessary condition for any descent optimization algorithm to provably find a global minimum from all initial conditions. It was shown in Venturi et al. (2018) that the empirical loss landscape of a shallow neural network with ReLU activation and squared loss has the descent path property if the size of training data is no larger than the width of the hidden layer, i.e., $n \leq m$. We now characterize a class of local minima of specific form in the following definition.

**Definition 2** (Cupped minima). *Given a function $f : \mathbb{R}^d \to \mathbb{R}$, we call $x \in \mathbb{R}^d$ a cupped minimum of $f$ if there are $\epsilon, \delta > 0$ such that for any $y$ in the $\delta$-neighborhood of $x$, we have $f(y) \geq f(x) + \epsilon \|y - x\|^2$. By a sub-optimal cupped minimum we mean a cupped minimum that is not a global minimum.*

Note that every cupped minimum is a local minimum, but not every local minimum is cupped (e.g., flat local minima are not curved downwards, and hence are not cupped). Also note that a function is not necessarily differentiable at its cupped minima. We study cupped minima of the loss function in equation 1. Note however that for any $\alpha > 0$, $F(\alpha w, v/\alpha) = F(w, v)$. Therefore, $F(\cdot, \cdot)$ has no cupped minima if both arguments are taken as variables. For that matter, when talking about cupped minima of $F$, we fix a $v$ and consider $F(\cdot, v)$ as a function of its first argument. Interestingly, existence of cupped minima for $F(\cdot, v)$ leads to violation of descent path property for $F(\cdot, \cdot)$ over both arguments, as shown in the following lemma. The proof is given in Appendix B.

**Lemma 1.** *Consider a shallow network with loss function $F$ in equation 1, and a pair of weights $(w, v)$. Suppose that $w$ is a sub-optimal cupped minimum of $F(\cdot, v)$, and that $w_r \neq 0$, for $r = 1, \ldots, m$. Then, $F(\cdot, \cdot)$ has no descent path $(w(t), v(t))$, initiated at $(w, v)$, to its global minima.*

## 3 MAIN RESULTS

The following theorem and corollary state the main results of the paper.

**Theorem 1.** *For any $d \geq 4$, $m \geq 8 + 4\lfloor 3/(d-3) \rfloor$, and $n \geq m + 2d - 2$, there exists a training set of size $n$ such that the empirical loss function $F$ of a shallow neural network of width $m$ has the following property. For any $m$-dimensional vector $v$, with $m/2$ number of positive and $m/2$ number of negative entries, $F(\cdot, v)$ has exponentially many sub-optimal cupped minima.*

The proof is constructive and is given in Section 4. In particular, we devise a training sequence $(X_1, y_1), \ldots, (X_n, y_n)$ such that for weights $(w, v)$ at the cupped minima, we have $\|v_r w_r\| = 1/\sqrt{m}$, for $r = 1, \ldots, m$. Moreover, $\|X_i\| \leq 1$, $|y_i| \leq 2$, and $|e_i| = 1$ for $i = 1, \ldots, n$ (cf. Remark 2).

According to Theorem 1, there are training sequences tailored to give rise to sub-optimal cupped minima. However, we wish to point that the existence of such cupped minima does not stem from measure-zero incidents like placement of several data points on a low dimensional plane. In fact, in view of Lemma 1, any path that starts from a cupped minimum and end up in a global minimum would have an uphill climb of at least $\epsilon$, for some $\epsilon > 0$. Since the loss surface is a continuous function of $(X_i, y_i)$, a small perturbation of $(X_i, y_i)$'s leads continuously to a small change in $F$. Therefore, for small enough perturbations of $(X_i, y_i)$, any path to the set of global minima would still witness a positive uphill-climb. Hence, the descent path property remains out of order, even when the training data is slightly perturbed. Based on the above intuition, we can establish the following corollary[2],

**Corollary 1.** *For any $d \geq 4$, $m \geq 8 + 4\lfloor 3/(d-3) \rfloor$, and $n \geq m + 2d - 2$; and when the inputs $X$ and labels $y$ are randomly drawn from independent normal distributions, there is a non-zero probability that $F(\cdot, \cdot)$ does not have the descent path property.*

---

[2]A formal proof is pretty tedious, and is not presented here.

Note that Corollary 1 does not imply Theorem 1, because a landscape could be cupped-min-free even if it the descent path property is not satisfied.

It was shown in Venturi et al. (2018) that $n \leq m$ is sufficient for the descent path property to hold. In contrast, Corollary 1 show that if $n \geq m + 2d - 2$, then the descent path property is not necessarily in effect. This leave a gap of size $2d - 2$ for the edge of over-parameterization required to guarantee the descent path property. We believe that this edge lies sharp at $n = m$. We conjecture a stronger version of Theorem 1, that cupped minima can emerge for training data sizes as small as $m = n - 4$.

**Conjecture 1.** *Statement of Theorem 1 holds for all $d \geq 4$, $m \geq 2d + 4$, and $n \geq m + 4$.*

See Remark 1 for insights into the possibility of this conjecture.

## 4 PROOF OF THE MAIN RESULT

In this section, we present the proof of Theorem 1 organized in a sequence of four subsections. We first present some preliminaries in Subsection 4.1. In Subsection 4.2, we introduce a geometric structure called "$(d, t, k)$-configuration", based on which we construct, in Subsection 4.3, the training set that gives rise to cupped minima in the loss landscape. Finally, in Subsection 4.4, we prove the existence of cupped mimima in the devised setting. In order to provide intuitions on the loss landscape at cupped minima and motivate our construction of the training set in Subsection 4.3, we make a short note on different types of cupped minima in Appendix A. We also defer the proofs of some lemmas from this section to appendices for improved readability.

### 4.1 PRELIMINARIES

Consider weights $(w, v)$ and let $(w', v')$ be another set of weights such that for $r = 1, \ldots, m$, $v_r$ and $v'_r$ have the same sign and $w'_r = (v_r/v'_r)w_r$. Then, $F(w, v) = F(w', v')$. Moreover, it is no difficult to see that $w$ is a cupped minimum of $F(\cdot, v)$ if and only if $w'$ is a cupped minimum of $F(\cdot, v')$. For this reason, it suffices to prove existence of cupped minima for a fixed vector $v$. Note also that where $w_r$'s are distinct, any permutation of the order of neurons would give rise to a new cupped minimum. Hence, existence of a cupped minimum for $F(\cdot, v)$ implies existence of exponentially many cupped minima for $F(\cdot, v)$.

We denote by $e_i = \hat{y}_i(w, v) - y_i$ the estimation error for input $X_i$. We let $u_d = [0, \ldots, 0, 1]^T$ be a $d$-dimensional vector with the last entry equal to one and all other entries equal to zero. For a region $\mathcal{P} \subseteq \mathbb{R}^d$, we denote its interior and and its convex-hull by $\text{int}(\mathcal{P})$ and $\text{Conv}(\mathcal{P})$, respectively. Assuming differentiability of $F$ at $w$, the partial derivatives of the loss function with respect to $w_r$, $r = 1, \ldots, m$, is as follows

$$\nabla_{w_r} F(w, v) = v_r \sum_{i=1}^{n} e_i X_i \mathbf{1}\left(w_r^T X_i \geq 0\right). \tag{2}$$

### 4.2 A GEOMETRIC CONFIGURATION

We introduce a geometric structure for sets of points in $\mathbb{R}^d$. This configuration will be used in Subsection 4.3 to construct a landscape with cupped minima.

**Definition 3** ($(d, t, k)$-Configuration). *Given integers $d, t, k \geq 0$ and disjoint sets $A$, $\tilde{A}$, $B$, and $\tilde{B}$ of points in $\mathbb{R}^d$, we say that $(A, \tilde{A}, B, \tilde{B})$ forms a $(d, t, k)$-configuration if the following properties are satisfied:*

(p1) *Each of $A$ and $B$ consists of $t$ points, and each of $\tilde{A}$ and $\tilde{B}$ consists of $k$ points.*

(p2) (A) *The convex hull of $A \bigcup \tilde{A}$ forms a polytope $\mathcal{P}_A$ that has exactly $t + k$ vertices. Equivalently, no point in $A \bigcup \tilde{A}$ is a convex combination of other points in $A \bigcup \tilde{A}$.*

    (B) *Similarly, the convex hull of $B \bigcup \tilde{B}$ forms a polytope $\mathcal{P}_B$ that has exactly $t + k$ vertices.*

(p3) *We have $0 \in \text{int}(\mathcal{P}_A)$ and $0 \in \text{int}(\mathcal{P}_B)$.*

(p4) *There exists a constant $\beta \in (0, 1)$ such that*

    (A) *For any $a \in A$, $\beta a$ lies on a facet of $\mathcal{P}_B$. We denote this facet by $S_B(a)$.*

(B) *For any $b \in B$, $\beta b$ lies on a facet of $\mathcal{P}_A$. We denote this facet by $S_A(b)$.*

(p5) (A) *For any $a \in A$, $S_B(a)$ is a $(d-1)$-dimensional simplex.*

(B) *For any $b \in B$, $S_A(b)$ is a $(d-1)$-dimensional simplex.*

(p6) (A) *For any $a \in A$, there exist scalars $\alpha_1, \ldots, \alpha_d \in (0,1)$ such that $a = \sum_{i=1}^{d} \alpha_i s_i(a)$, where $s_1(a), \ldots, s_d(a)$ are the vertices of simplex $S_B(a)$.*

(B) *For any $b \in B$, there exist scalars $\alpha_1, \ldots, \alpha_d \in (0,1)$ such that $b = \sum_{i=1}^{d} \alpha_i s_i(b)$, where $s_1(b), \ldots, s_d(b)$ are the vertices of simplex $S_A(b)$.*

(p7) (A) *For any pair $a$ and $a'$ of distinct points in $A$, we have $S_B(a) \neq S_B(a')$. Moreover, letting $\mathcal{H}_a$ be the hyperplane that contains $S_B(a)$, $a$ and $a''$ lie on opposite sides of $\mathcal{H}_a$, for all $a'' \in A \bigcup \tilde{A}$ with $a'' \neq a$.*

(B) *For any pair $b$ and $b'$ of distinct points in $B$, we have $S_A(b) \neq S_A(b')$. Moreover, letting $\mathcal{H}_b$ be the hyperplane that contains $S_A(b)$, $b$ and $b''$ lie on opposite sides of $\mathcal{H}_b$, for all $b'' \in B \bigcup \tilde{B}$ with $b'' \neq b$.*

(p8) *Consider a $2t \times 2t$ matrix $M$ whose rows and columns are associated to points $p \in A \bigcup B$ and $q \in A \bigcup B$, and whose entries are as follows*

$$M_{pq} = \begin{cases} d(p, \mathcal{H}_q), & \text{if } p = q, \ OR \ p \in A \text{ and } q \in B, \ OR \ p \in B \text{ and } q \in A, \\ 0, & \text{otherwise}, \end{cases} \tag{3}$$

*where $\mathcal{H}_q$ is the hyperplane define in Property (p7), and $d(\cdot, \cdot)$ is the euclidean distance. The property requires $M$ to be full-rank.*

Among the above properties, the most difficult of all is Property (p4), and the requirement that it involves the same $\beta$ for all points in $A \bigcup B$. In fact, elimination of Property (p4) gives rise to trivial configurations. [3]

In the two dimensional space, for any $t \geq 4$ there exists a $(2, t, 0)$-configuration of the form illustrated in Fig. 2. In the following proposition, we generalize this observation to higher dimensions.

**Proposition 1.** *For any $d \geq 2$ and $t \geq 4$, there exists a $(d, t, d-2)$-configuration.*

The proof is constructive, and is given in Appendix C. We conjecture that there also exist $(d, t, 0)$-configurations.

**Conjecture 2.** *For any $d \geq 2$ and $t \geq 2d$, there exists a $(d, t, 0)$-configuration.*

**Remark 1.** *Using a configuration given by Conjecture 2 instead of the configuration from Proposition 1 in the construction and the proof that follow, we obtain a proof for Conjecture 1. In this view, establishing Conjecture 2 would also resolve Conjecture 1.*

### 4.3 Constructing a landscape with cupped minima

Here we present a set of training data $(X_1, y_1), \ldots, (X_n, y_n)$ and a set of wights $(w_1, v_1), \ldots, (w_m, v_m)$ such that the empirical loss surface corresponding to $(X_1, y_1), \ldots, (X_n, y_n)$ has a sub-optimal cupped minimum at $(w_1, v_1), \ldots, (w_m, v_m)$. Without loss of generality, we assume $n = m + 2d - 2$. Extension to larger values of $n$ is straightforward via replication.

Let $(A, \tilde{A}, B, \tilde{B})$ be a $(d-1, m/2, d-3)$-configuration in the $(d-1)$-dimensional space, as in Proposition 1. In view of Property (p3), let $\epsilon_0 > 0$ be such that $\mathcal{P}_A$ and $\mathcal{P}_B$ contain the $\epsilon_0$-neighborhood of 0. Let

$$\xi \triangleq \frac{1}{\epsilon_0} \left( \| \sum_{a \in A \bigcup \tilde{A}} a \| + \| \sum_{b \in B \bigcup \tilde{B}} b \| \right) + n + \frac{1}{\beta}. \tag{4}$$

We proceed by introducing the data points $X_1, \ldots, X_n$. An illustration of these data points in the three dimensional space is shown in Fig. 3.

---

[3]e.g., for $A$ and $\tilde{A}$ chosen uniformly at random over the unit sphere, and letting $(B, \tilde{B})$ be a small rotation of $(A, \tilde{A})$ around an arbitrary axis, it is not difficult to see that all properties except (p4) would be satisfied with high probability.

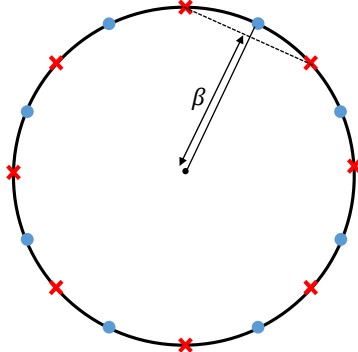

Figure 2: A $(2, t, 0)$-configuration with $t = 8$. The blue dots show the points in $A$ and red crosses are the points in $B$.

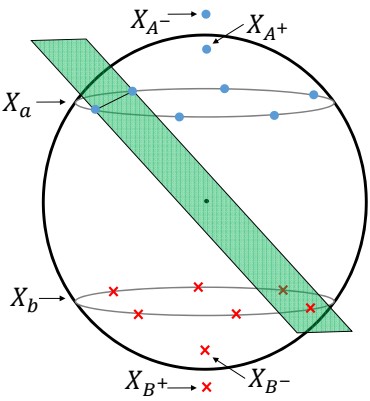

Figure 3: Illustration of data points $X_1, \ldots, X_n$ for $d = 3$ and $m = 12$.

**Data points $X$:** We consider a total number of $n = m + 2d - 2$ data points as follows.

- For each $a \in A \bigcup \tilde{A}$, we consider a new data point $X_a$ as follows. Let $[z_1, \ldots, z_{d-1}] \in \mathbb{R}^{d-1}$ be the representation of $a$ in the Cartesian coordinates. We let $X_a = [z_1, \ldots, z_{d-1}, 1]$.

- For each $b \in B \bigcup \tilde{B}$, we consider a new data point $X_b$ as follows. Let $[z_1, \ldots, z_{d-1}] \in \mathbb{R}^{d-1}$ be the representation of $b$ in the Cartesian coordinates. We let $X_b = -[z_1, \ldots, z_{d-1}, 1]$.

- We consider two extra points $X_{A^+}, X_{A^-}, X_{B^+}$, and $X_{B^-}$ as follows. We let $X_{A^-} \triangleq \xi u_d$, $X_{B^-} \triangleq -\xi u_d$, and

$$X_{A^+} \triangleq X_{A^-} - \sum_{a \in A \bigcup \tilde{A}} X_a + (1/\beta - 1) u_d \tag{5}$$

$$X_{B^+} \triangleq X_{B^-} + \sum_{b \in B \bigcup \tilde{B}} X_b + (1/\beta - 1) u_d \tag{6}$$

where $u_d = [0, \ldots, 0, 1]^T$, and $\xi$ and $\beta$ are defined in equation 4 and Property (p4), respectively.

**Weights at cupped minimum:** We associate each of $m$ neurons to a point $p$ in $A \bigcup B$, in a one-one manner; and denote the vector of input weights and the output weight of that neuron by $w_p$ and $v_p$, respectively. These weights are chosen as follows:

- For each $a \in A$, we let $v_a = -1/\sqrt{m}$.
- For each $b \in B$, we let $v_b = 1/\sqrt{m}$.
- For each $a \in A$, consider the facet $S_B(a)$ defined in Property (p4), and let $s_1(a), \ldots, s_{d-1}(a)$ be the vertices of $S_B(a)$ (as in Property (p6)). We let $w_a$ be the unique vector such that $\|w_a\| = 1$ and

$$w_a^T X_{s_i(a)} = 0, \qquad i = 1, \ldots, d-1, \tag{7}$$

$$w_a^T u_d < 0. \tag{8}$$

- For each $b \in B$, consider the facet $S_A(b)$ and let $s_1(b), \ldots, s_{d-1}(b)$ be the vertices of $S_A(b)$. We let $w_b$ be the unique vector such that $\|w_b\| = 1$ and

$$w_b^T X_{s_i(b)} = 0, \qquad i = 1, \ldots, d-1, \tag{9}$$

$$w_b^T u_d > 0. \tag{10}$$

**Labels $y$:** Having determined the data points $X$ and the weights $(w, v)$, the output $\hat{y}(w, v)$ of the network is determined for all input $X$. In the following, we choose the true labels $y$ to obtain a desired error $e = \hat{y} - y$ for each input data. In particular:

- For each $a \in A \bigcup \tilde{A}$, we associate to $X_a$ a label $y_a$ so that $e_a \triangleq \hat{y}_a(w, v) - y_a = 1$.

- For each $b \in B \bigcup \tilde{B}$, we associate to $X_b$ a label $y_b$ so that $e_b \triangleq \hat{y}_b(w, v) - y_b = -1$.

- We choose the labels associated to $X_{A^+}$, $X_{A^-}$, $X_{B^+}$, and $X_{B^-}$ such that $e_{A^+} = e_{B^+} = 1$ and $e_{A^-} = e_{B^-} = -1$.

This completes the description of the training set. As shown in in Soudry and Carmon (2016), there exist weights that achieve zero loss if $m > 4\lceil n/(2d - 2)\rceil$. In our case, $n = m + 2d - 2$, and its easy to check that $m > 4\lceil (m + 2d - 2)/(2d - 2)\rceil$ for all $d \geq 4$ and $m \geq 8 + 4\lfloor 3/(d - 3)\rfloor$. It follows that in our setting the global optimum has zero loss, showing that the above $(w, v)$ is sub-optimal.

**Remark 2.** *In the above construction, the norms of inputs vectors may be very large. If we scale down the inputs such that $\|X_i\| \leq 1/\sqrt{m}$ for $i = 1, \ldots, n$, and modify the corresponding labels $y_i$ such that $e_i$ remains unchanged, then a same set of weights will still be a cupped minimum for the landscape defined in terms of new $(X_i, y_i)$'s. Moreover, for this setting, it is easy to check that $\|w_r\| = 1$, $\|e_i\| = 1$, and $|y_i| \leq 2$.*

## 4.4 PROVING THE CUPPED MINIMA PROPERTY

Let

$$\delta \triangleq \min \left\{ \frac{|w_r^T X_i|}{\|X_i\|} \ \Big| \ w_r^T X_i \neq 0, \ r = 1, \ldots, m, \ i = 1, \ldots, n \right\}. \tag{11}$$

Then, $\delta > 0$. For any $\theta \in \mathbb{R}^{md}$ with $\|\theta\| = 1$ and for any $t \in [0, \delta]$, let

$$F_\theta(t) = F(w + \theta t, v). \tag{12}$$

We show that there is an $\epsilon > 0$ such that for any unit-norm $\theta$ and any $t \in [0, \delta]$,

$$F_\theta(t) \geq F_\theta(0) + \epsilon t^2. \tag{13}$$

**Lemma 2.** *For any $\theta \in \mathbb{R}^{md}$ with $\|\theta\| = 1$, $F_\theta(\cdot)$ is a quadratic and convex function over $[0, \delta]$.*

The proof is give in Appendix D and relies on the fact that neuron activations do not alter at $w + \theta t$ for $t \in [0, \delta]$. Consider now the following $m$-dimensional subspace of $\mathbb{R}^{md}$

$$H_w \triangleq \left\{ \begin{bmatrix} \alpha_1 w_1 \\ \vdots \\ \alpha_m w_m \end{bmatrix} \ \Big| \ \alpha_1, \ldots, \alpha_m \in \mathbb{R} \right\}. \tag{14}$$

For any $\theta \in \mathbb{R}^{md}$, let $\theta^\parallel$ and $\theta^\perp$ be the orthogonal projections of $\theta$ on $H_w$ and $H_w^\perp$, respectively. Then, $\theta = \theta^\parallel + \theta^\perp$. In order to establish equation 13, we need lower bounds on $F_\theta'(0)$ and $F_\theta''$, which we derive in the next two lemmas.

**Lemma 3.** *There exists $\mu > 0$ such that for any $\theta \in \mathbb{R}^{md}$ with $\|\theta\| = 1$, we have*

$$\left. \frac{dF_\theta(t)}{d^+t} \right|_{t=0} \geq \mu \|\theta^\perp\|. \tag{15}$$

The proof is given in Appendix E, and relies in a subtle way on the choice of data points in subsection 4.3. We now bound the curvature of $F_\theta$.

**Lemma 4.** *There exist constants $\eta_1, \eta_2 > 0$ such that for any $\theta \in \mathbb{R}^{md}$ with $\|\theta\| = 1$, and for any $t \in (0, \delta)$,*

$$\frac{d^2 F_\theta(t)}{dt^2} \geq \max \left( 2\eta_1 \|\theta^\parallel\|^2 - 2\eta_2 \|\theta^\perp\|, \ 0 \right). \tag{16}$$

The proof is given in Appendix F, and relies on Property (p8) of the underlying configuration.

Consider now the second order polynomial $p(x) = \eta_1 \delta (1 - x^2) - \eta_2 \delta x - \mu x$, where $\mu$, $\eta_1$, and $\eta_2$ are the constants in Lemmas 3 and 4. Since $p(0) > 0$ and $p(1) < 0$, the polynomial $p$ has exactly one root in the interval $(0, 1)$, which we denote by $x_0$. Let

$$\epsilon \triangleq \mu x_0 / \delta. \tag{17}$$

**Lemma 5.** *For any $\theta \in \mathbb{R}^{md}$ with $\|\theta\| = 1$, and any $t \in [0, \delta]$, we have $F_\theta(t) \geq F_\theta(0) + \epsilon t^2$.*

This lemma is a simple consequence of Lemmas 3 and 4, and its proof is given in Appendix G. It follows from Lemma 5 that for any $w'$ in the $\delta$-neighborhood of $w$, we have $F(w', v) \geq F(w, v) + \epsilon \|w' - w\|^2$. This shows that $w$ is a cupped minimum for $F(\cdot, v)$, and completes the proof of Theorem 1.

## 5    DISCUSSION

The guaranteed existence of descent paths in shallow networks of ReLU neurons was previously established (Venturi et al., 2018), given an over-parameterization of $m \geq n$ (where $m$ and $n$ are the number of neurons and the size of training data, respectively). This left an uncertainty gap of $2n/d < m < n$, where zero empirical risk is known to be achievable (for $m \geq 2n/d$) (Soudry and Hoffer, 2017), but the existence of descent paths was in question. In this work, we have tightened this uncertainty gap to $n - 2d + 2 < m < n$, by proving that for any $m \leq n - 2d + 2$, there are input data and initial weights for which a descent path does not exist. This conclusion we reach by establishing the existence of cupped minima for $m \leq n - 2d + 2$, and for the right choice of input data.

Compared to similar existing results for other activation functions, our results suggest that the edge $m \approx n$ of over-parameterization required for elimination of sub-optimal cupped minima in ReLU networks is much higher than that of networks with quadratic activation functions, $m \approx \sqrt{2n}$ (Du and Lee, 2018), and linear activation functions, $m \approx n/d$ (Kawaguchi, 2016), and is almost as high as general continuous activation functions of any form, $m \leq n$ (Venturi et al., 2018).

Non-existence of spurious local minima and the decent path property favor the convergence of decent optimization methods like GD. However, for different variants of noisy GD, like SGD and Langevin dynamics, it is quite common for the empirical loss to fluctuate during the training. Nevertheless, for theoretical analysis purposes it usually helps to take the noise away, for example by tending the step size to zero. The resulting GD, which always follows a descent path, is usually easier to analyze and can also help to study the SGD dynamics. On the other hand, from a practical view, convergence of SGD in local-min-free landscapes is well-studied.

Aside from addressing Conjecture 2, there remain several open problems, which we review next. As an important direction for future research, it would be interesting if one could obtain bounds on the probability of existence of cupped minima over random data sets, underneath the edge of over-parameterization. In particular, we showed in Corollary 1 that this probability is non-zero, however we gave no clue on either the size or scaling of this probability. Another class of problems concerns basins of local minima, and how they affect dynamics of first order optimization algorithms. As a step toward this goal, one might characterize the true over-parameterization regime in which the basins of poor local minima have considerable volume.

Among other directions are extensions of our results to deep ReLU networks, shallow non-ReLU networks, and shallow ReLU networks under loss functions more general than the squared loss.

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

# APPENDICES

## A    DIFFERENT TYPES OF CUPPED MINIMA IN TERMS OF DIFFERENTIABILITY

Here we discuss different types of cupped minima and provide elementary intuitions on the loss landscape at cupped minima.

We first characterize curvature of the loss function at differentiable points. For $r = 1, \ldots, m$, let $J_r$ be an $n \times n$ diagonal matrix whose $(i, i)$ entry equals $\mathbf{1}\left(w_r^T X_i \geq 0\right)$. Let $G$ be an $n \times md$ matrix of the form:

$$G = \left[ \begin{array}{ccc} v_1 J_1 \mathbf{X}^T & \cdots & v_m J_m \mathbf{X}^T \end{array} \right], \tag{18}$$

where $\mathbf{X}$ is a $d \times n$ matrix that has $X_i$ in its $i$-th column. If $F$ is differentiable at $(w, v)$, its gradient is $\nabla_w F(w, v) = G^T [e_1, \ldots, e_m]^T$, where $e_i = \hat{y}_i - y_i$ is the output error for input $X_i$. Moreover, if $F$ is differentiable at $(w, v)$, its Hessian with respect to $w$ equals

$$\nabla_w^2 F(w, v) = G^T G. \tag{19}$$

We classify cupped minima into three categories in terms of differentiability. Specifically, for a cupped minimum $w$ of $F(\cdot, v)$, we consider three cases:

Type 1)  $F$ is differentiable at $w$.

Type 2)  $F(w + \theta t, v)$ as a function of $t$ is non-differentiable at $t = 0$, for all $\theta \in \mathbb{R}^{md}$.

Type 3)  There are $\theta_1, \theta_2 \in \mathbb{R}^{md}$ such that $F(w + \theta_1 t, v)$ is differentiable at $t = 0$, while $F(w + \theta_2 t, v)$ is non-differentiable at $t = 0$.

Fig. 4 illustrates examples of loss surface at the above three types of cupped minima. We now argue that the first two types are not possible in the loss landscape of shallow networks.

If $F(\cdot, v)$ is differentiable at $w$, its Hessian given in equation 19 equals $G^T G$. Since $G$ is an $n \times md$ matrix, assuming $md > n$, $G^T G$ would have zero eigenvalues. Therefore, $w$ cannot be a cupped minimum of $F(\cdot, v)$. It follows that there exists no differentiable cupped minimum (nor saddle point) if $md > n$.

For non-differentiable points, note that $F(w + wt, v)$ as a function of $t$ is a differentiable quadratic function. This is because the output scales proportionally with $w$. Therefore, cupped minima of the second type are not possible, as well. In the same spirit, it can be shown that $F(w + \theta t, v)$ is differentiable as a function of $t$, if $\theta \in \mathbb{R}^{md}$ belongs to the $m$ dimensional subspace $H_w$ defined in equation 14.

For the above reasons, all cupped minima, if any, are of the third type. Therefore in the construction of Subsection 4.3, we introduce a training set and a pair of weights $(w, v)$ such that $w$ is a cupped minimum of $F(\cdot, v)$, and $F(w + \theta t, v)$ is differentiable in $t$ only for $\theta \in H_w$.

## B    PROOF OF LEMMA 1

If $v$ has a zero entry, $v_r = 0$ for some $r$, then $F(\cdot, v)$ is a constant function with respect to $w_r$, and thereby has no cupped minima. Therefore, we assume that $v$ has no zero entries. Let

$$t_0 = \inf \left\{ t \mid \exists r, \ v_r(t) = 0 \right\}.$$

For $t > 0$, we define

$$\tilde{w}(t) = \left[ \begin{array}{c} \left| v_1(t)/v_1 \right| \times w_1(t) \\ \vdots \\ \left| v_m(t)/v_m \right| \times w_m(t) \end{array} \right]. \tag{20}$$

Let

$$t_1 = \inf \left\{ t \mid \tilde{w}(t) \neq w \right\}.$$

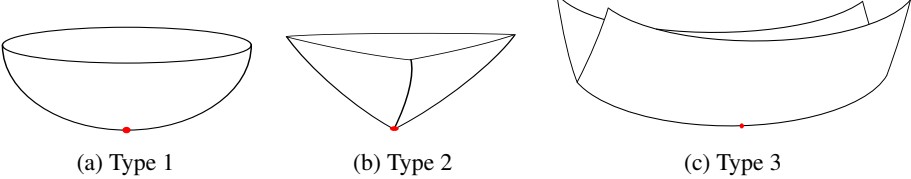

(a) Type 1           (b) Type 2           (c) Type 3

Figure 4: Different types of cupped minima in terms of differentiability, discussed in Appendix A.

We show that $t_1 < t_0$. If $t_0 < \infty$, then continuity of $v(t)$ implies that $v_r(t_0) = 0$, for some $r \leq m$. Therefore, $\tilde{w}_r(t_0) = 0 \neq w_r$. It then follows from the continuity of $\tilde{w}_r(\cdot)$ that there is an $\epsilon > 0$ such that $\tilde{w}_r(t_0 - \epsilon) \neq w_r$. Consequently, $t_1 < t_0$.

The inequality $t_1 < t_0$ implies that there is an $\epsilon > 0$ such that $\tilde{w}_r(t_1 + \epsilon) \neq w$, and for any $t \in [0, t_1 + \epsilon]$, we have $\mathrm{sgn}(v_r(t)) = \mathrm{sgn}(v_r)$, $r = 1, \ldots, m$. Therefore, for any $t \in [0, t_1 + \epsilon]$, we have $F(\tilde{w}(t), v) = F(w(t), v(t))$. Since $w$ is a cupped minimum of $F(\cdot, v)$, there is an $s \in [t_1, t_1 + \epsilon]$ such that $F(\tilde{w}(t), v) > F(w, v)$. This shows that $(w(t), v(t))$ cannot be a descent path for $F(\cdot, \cdot)$, and completes the proof of Lemma 1.

## C  PROOF OF PROPOSITION 1

For $i = 3, \ldots, d$, let

$$c_i \triangleq \frac{1}{i - 1 + 2\cos(\pi/t)}. \tag{21}$$

Fix a constant

$$\tilde{c} \triangleq \prod_{i=3}^{d} (1 + c_i) = \prod_{i=3}^{d} \frac{i + 2\cos(\pi/t)}{i - 1 + 2\cos(\pi/t)} = \frac{d + 2\cos(\pi/t)}{2 + 2\cos(\pi/t)}. \tag{22}$$

Let $\gamma$ be a uniform random variable

$$\gamma \sim \mathrm{unif}\left(\frac{\cos(\pi/t) - \cos(2\pi/t)}{4\tilde{c}}, \frac{\cos(\pi/t) - \cos(2\pi/t)}{2\tilde{c}}\right). \tag{23}$$

We take a sample from the above distribution and fix a $\gamma$ for the rest of the proof.

We now introduce the points in the configuration. For $i = 0, \ldots, t - 1$, we consider points $a^i \in A$ and $b^i \in B$ as follows

$$a^i = \left[\cos\left(\frac{2\pi(i - 1/2)}{t}\right), \sin\left(\frac{2\pi(i - 1/2)}{t}\right), \gamma c_3, \gamma c_4, \ldots, \gamma_d c_d\right], \tag{24}$$

$$b^i = \left[\cos\left(\frac{2\pi i}{t}\right), \sin\left(\frac{2\pi i}{t}\right), -\gamma c_3, -\gamma c_4, \ldots, -\gamma_d c_d\right]. \tag{25}$$

For $i = 3, \ldots, d$, we consider points $\tilde{a}^i \in \tilde{A}$ and $\tilde{b}^i \in \tilde{B}$ as follows

$$\tilde{a}^i = \Big[\underbrace{0, \ldots, 0}_{i-1}, -1, c_{i+1}, \ldots, c_d\Big], \tag{26}$$

$$\tilde{b}^i = \Big[\underbrace{0, \ldots, 0}_{i-1}, 1, -c_{i+1}, \ldots, -c_d\Big]. \tag{27}$$

Fig. 5 shows an illustration of these points for $d = 3$.

We proceed by verifying Properties (p1)–(p8).

**Property** (p1): Property (p1) is straightforward from the above construction.

**Property** (p2): Considering only the first two coordinates, it is easy to see that no point of $A$ is a convex combination of other points in $A \bigcup \tilde{A}$. Also note that for $i = 3, \ldots, d$, $\tilde{a}^i$ is the only point

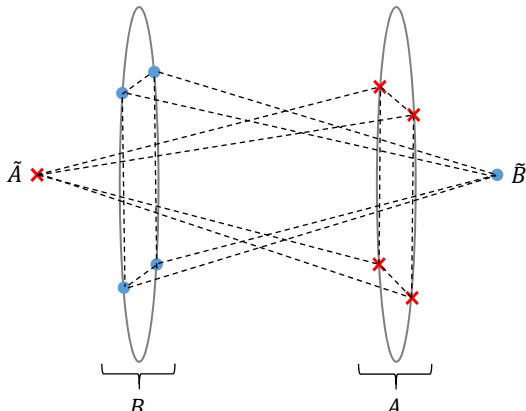

Figure 5: An illustration of the points in equation 24–equation 27 for $d = 3$ and $t = 4$. The red crosses indicate the points in $A \bigcup \tilde{A}$ and the blue dots correspond to the points in $B \bigcup \tilde{B}$.

in $A \bigcup \tilde{A}$ whose $i$-th entry is negative. Therefore, no point in $\tilde{A}$ can be represented as a convex combination of other points in $A \bigcup \tilde{A}$. A similar argument works for Part (B) of Property (p2).

**Property** (p3): Let

$$\alpha_i = \frac{1}{\gamma t}, \qquad i = 1, \ldots, t, \tag{28}$$

$$\tilde{\alpha}_i = (c_3 + 1) \cdots (c_{i-1} + 1)c_i, \qquad i = 3, \ldots, d. \tag{29}$$

A simple induction on $j$ shows that

$$\sum_{i=3}^{j-1} \tilde{\alpha}_i = \frac{\tilde{\alpha}_j}{c_j} - 1. \tag{30}$$

for $j = 4, \ldots, d$. We show that

$$\frac{1}{\sum_{i=0}^{t-1} \alpha_i + \sum_{i=3}^{d} \tilde{\alpha}_i} \left( \sum_{i=0}^{t-1} \alpha_i a^i + \sum_{i=3}^{d} \tilde{\alpha}_i \tilde{a}^i \right) = 0. \tag{31}$$

For the first two coordinates, equation 31 is easy. Let $a_j^i$ and $\tilde{a}_j^i$ be the $j$-th entries of $a^i$ and $\tilde{a}^i$, respectively. For the $j$-th coordinate, $j = 3, \ldots, d$, we then have

$$\sum_{i=0}^{t-1} \alpha_i a_j^i + \sum_{i=3}^{d} \tilde{\alpha}_i \tilde{a}_j^i = t \times \frac{1}{t\gamma} \times \gamma c_j + c_j \sum_{i=3}^{j-1} \tilde{\alpha}_i - \tilde{\alpha}_j$$

$$= c_j + c_j \left( \frac{\tilde{\alpha}_j}{c_j} - 1 \right) - \tilde{\alpha}_j$$

$$= 0,$$

where the second equality is due to equation 30. This establishes equation 31. It then follows from equation 31 that 0 is a convex combination of points in $A \bigcup \tilde{A}$. Consequently, $0 \in \text{int}(\mathcal{P}_A)$. A similar argument shows that $0 \in \text{int}(\mathcal{P}_B)$.

**Properties** (p4)–(p7): We only prove Part (B) for each of these properties; as similar proofs work also for Part (A)'s. Moreover, because of the rotational symmetry of $A$ and $B$ in the first two coordinates, it suffices to prove of the Properties (p4)–(p7) only for $b^0$.

**Properties** (p4) **and** (p5): For the $j$-th coordinate, $j = 3, \ldots, d$, we have

$$
\begin{aligned}
a_j^0 + a_j^1 + \gamma \sum_{i=3}^{d} \tilde{a}_j^i &= \gamma c_j + \gamma c_j + \gamma \sum_{i=3}^{j-1} c_j - \gamma \\
&= \gamma \big( (j-1) c_j - 1 \big) \\
&= \gamma \left( (j-1) \frac{1}{j - 1 + 2\cos(\pi/t)} - 1 \right) \\
&= \frac{-2\gamma \cos(\pi/t)}{j - 1 + 2\cos(\pi/t)} \\
&= -\big( 2\cos(\pi/t) \big) \gamma c_j \\
&= 2\cos(\pi/t) \, b_j^0,
\end{aligned}
\tag{32}
$$

where $b_j^0$ is the $j$-th entry of $b^0$ defined in equation 25. Similarly, for the first two coordinates, we have:

$$
\begin{aligned}
a_1^0 + a_1^1 + \gamma \sum_{i=3}^{d} \tilde{a}_1^i &= \cos(\pi/t) + \cos(\pi/t) = 2\cos(\pi/t) \, b_1^0, \\
a_2^0 + a_2^1 + \gamma \sum_{i=3}^{d} \tilde{a}_2^i &= \sin(\pi/t) - \sin(\pi/t) = 0 = 2\cos(\pi/t) \, b_2^0.
\end{aligned}
\tag{33}
$$

It then follows from equation 32 and equation 33 that

$$
b^0 = \frac{1}{2\cos(\pi/t)} \left( a^0 + a^1 + \gamma \sum_{i=3}^{d} \tilde{a}_i \right).
\tag{34}
$$

Let

$$
\beta \triangleq \frac{2\cos(\pi/t)}{(d-2)\gamma + 2}.
\tag{35}
$$

Then, from equation 34,

$$
\beta b^0 = \frac{1}{(d-2)\gamma + 2} \left( a^0 + a^1 + \gamma \sum_{i=3}^{d} \tilde{a}_i \right).
\tag{36}
$$

Therefore, $\beta b^0$ is a convex combination of $a^0, a^1, \tilde{a}^3, \ldots, \tilde{a}^d$, and therefore lies on the simplex $S$ that has $a^0, a^1, \tilde{a}^3, \ldots, \tilde{a}^d$ as its vertices. Next, we show that $S$ is a facet of $\mathcal{P}_A$.

Consider a vector $z \in \mathbb{R}^d$ with entries

$$
\begin{aligned}
z_1 &= \frac{(\tilde{c} - 1)\gamma + 1}{\cos(\pi/t)}, \\
z_2 &= 0, \\
z_i &= -\prod_{j=i+1}^{d} (c_j + 1), \qquad i = 3, \ldots, d-1, \\
z_d &= -1.
\end{aligned}
\tag{37}
$$

Then, a simple backward induction, with base case $i = d$, shows that for $i = 3, \ldots, d$, we have $\sum_{j=i+1}^{d} c_j z_j = z_i + 1$. In the same vein,

$$
\sum_{j=3}^{d} c_j z_j = -\tilde{c} + 1.
\tag{38}
$$

It follows that for $i = 3, \ldots, d$,

$$
z^T \tilde{a}^i = \sum_{j=i+1}^{d} z_j c_j - z_i = 1.
\tag{39}
$$

Moreover, for $i = 0, 1$,

$$
\begin{aligned}
z^T a^i &= \sum_{j=3}^{d} z_j \gamma c_j \;+\; z_1 \cos(\pi/t) \\
&= \gamma(-\tilde{c} + 1) \;+\; z_1 \cos(\pi/t) \\
&= -\gamma(\tilde{c} - 1) \;+\; \frac{(\tilde{c} - 1)\gamma + 1}{\cos(\pi/t)} \cos(\pi/t) \\
&= 1,
\end{aligned}
\tag{40}
$$

where the second equality is due to equation 38. Let $\mathcal{H}$ be the hyperplane that passes through $a^0, a^1, \tilde{a}^3, \ldots, \tilde{a}^d$. It follows from equation 39 and equation 40 that for any $p \in \{a^0, a^1, \tilde{a}^3, \ldots, \tilde{a}^d\}$, we have $z^T p = 1$. Therefore, $z$ is orthogonal to $\mathcal{H}$. Consequently,

$$
\mathcal{H} = \{x \in \mathbb{R}^d \mid z^T x = 1\}.
\tag{41}
$$

For $i = 2, \ldots, t - 1$, we have

$$
\begin{aligned}
z^T a^i &= z^T (a^i - a^0) \;+\; z^T a^0 \\
&= z^T (a^i - a^0) \;+\; 1 \\
&= z_1 \big( \cos(2\pi(i - 1/2)/t) \;-\; \cos(\pi/t) \big) \;+\; 1 \\
&< 1,
\end{aligned}
\tag{42}
$$

where the second equality is due to equation 40, and the inequality is because $z_1 > 0$ and $\cos(2\pi(i - 1/2)/t) - \cos(\pi/t) < 0$. It follows from equation 41 and equation 42 that all points of $A \backslash \{a^0, a^1\}$ lie on one side of $\mathcal{H}$, while all points of $\tilde{A} \bigcup \{a^0, a^1\}$ lie on $\mathcal{H}$. Then, $\mathcal{H}$ is a tangent hyperplane to $\mathcal{P}_A$. Thus, the simplex $S$ is a facet of $\mathcal{P}_A$. This completes the proofs of Properties (p4) and (p5). Moreover, from the definition $\mathcal{H}_b$ in Property (p7), we have

$$
\mathcal{H}_{b^0} = \mathcal{H}.
\tag{43}
$$

**Property (p6):** Since $t \geq 4$, we have $2\cos(\pi/2) > 1$. Moreover, recall from equation 23 that $\gamma < 1$. Property (p6) then follows from equation 34 and the fact that $S$ is a facet of $\mathcal{P}_A$.

**Property (p7):** For $i = 0, \ldots, t - 1$,

$$
\begin{aligned}
z^T b^i &= z^T (b^i - a^0) \;+\; z^T a^0 \\
&= z_1 \big( \cos(2\pi i/t) - \cos(\pi/t) \big) \;-\; 2 \sum_{j=3}^{d} z_j \gamma c_j \;+\; 1 \\
&= z_1 \big( \cos(2\pi i/t) - \cos(\pi/t) \big) \;+\; 2\gamma(\tilde{c} - 1) \;+\; 1.
\end{aligned}
\tag{44}
$$

where the second equality is due to equation 40 and definitions of $a^0$ and $b^i$, and the third equality is from equation 38. It follows that

$$
z^T b^0 = z_1 \big( 1 - \cos(\pi/t) \big) \;+\; 2\gamma(\tilde{c} - 1) \;+\; 1 > 1,
\tag{45}
$$

where the inequality is because the first two terms on the left hand side of the inequality are positive. In the same vein, for $i = 1, \ldots, t - 1$

$$
\begin{aligned}
z^T b^i &= z_1 \big( \cos(2\pi i/t) - \cos(\pi/t) \big) \;+\; 2\gamma(\tilde{c} - 1) \;+\; 1 \\
&< z_1 \big( \cos(2\pi/t) - \cos(\pi/t) \big) \;+\; 2\gamma\tilde{c} \;+\; 1 \\
&< \big( \cos(2\pi/t) - \cos(\pi/t) \big) \;+\; 2\gamma\tilde{c} \;+\; 1 \\
&\leq \big( \cos(2\pi/t) - \cos(\pi/t) \big) \;+\; \big( \cos(\pi/t) - \cos(2\pi/t) \big) \;+\; 1 \\
&= 1,
\end{aligned}
\tag{46}
$$

where the second inequality follows from the definition of $z_1$ in equation 37 and the fact that $z_1 > 1$, and the third inequality is from the definition of $\gamma$ in equation 23 and the fact that $2\gamma\tilde{c} \leq$

$\cos(\pi/t) - \cos(2\pi/t)$. It then follows from equation 41, equation 45, and equation 46 that $b^0$ and $b^i$ lie on opposite sides of $\mathcal{H}$, for $i = 1, \ldots, t-1$.

On the other hand, since $\tilde{b}^i = -\tilde{a}^i$, for $i = 3, \ldots, d$, it follows from equation 39 that $z^T \tilde{b}^i = -z^T \tilde{a}^i = -1 < 1$. Therefore, for $i = 1, \ldots, 3$, $\tilde{b}^i$ and $b^0$ lie on opposite sides of $\mathcal{H}$. This completes the proof of Property (p7).

**Property** (p8): We have

$$
\begin{aligned}
d\big(b^0, \mathcal{H}_{b^0}\big) &= \frac{1}{\|z\|}\big(z^T b^0 - 1\big) \\
&= \frac{z_1\big(1 - \cos(\pi/t)\big) + 2\gamma(\tilde{c} - 1)}{\|z\|} \\
&= \frac{z_1}{\|z\|}\left(\big(1 - \cos(\pi/t)\big) + \frac{\gamma}{z_1}2(\tilde{c} - 1)\right),
\end{aligned}
\tag{47}
$$

where the first equality is due to equation 43 and equation 41, and the second equality follows from the equality in equation 45. Similarly, from equation 42, we have for $i = 0, \ldots, t-1$

$$
\begin{aligned}
d\big(a^i, \mathcal{H}_{b^0}\big) &= \frac{1}{\|z\|}\big(z^T a^i - 1\big) \\
&= \frac{z_1}{\|z\|}\Big(\cos\big(2\pi(i - 1/2)/t\big) - \cos(\pi/t)\Big).
\end{aligned}
\tag{48}
$$

For $i, j \in \{0, \ldots, t-1\}$, let $\hat{m}_{a^i, b^j} \triangleq \cos\big(2\pi(i - j - 1/2)/t\big) - \cos(\pi/t)$ and $\hat{m}_{b^i, a^j} \triangleq \cos\big(2\pi(i - j + 1/2)/t\big) - \cos(\pi/t)$. Then, it follows from *equation* 48 and rotational symmetry of $A$ and $B$ in the first two coordinates that for $i, j \in \{0, \ldots, t-1\}$,

$$
d\big(a^i, \mathcal{H}_{b^j}\big) = \frac{z_1}{\|z\|}\hat{m}_{a^i, b^j}, \qquad d\big(b^i, \mathcal{H}_{a^j}\big) = \frac{z_1}{\|z\|}\hat{m}_{b^i, a^j}.
\tag{49}
$$

Note that for any $p, q \in A \bigcup B$, $\hat{m}_{p,q}$ is a constant independent of the value of $\gamma$. Let $\hat{M}$ be a $2t \times 2t$ matrix, with entries

$$
\hat{M}_{pq} = \begin{cases} 1 - \cos(\pi/t), & \text{if } p = q, \\ \hat{m}_{p,q}, & \text{if } p \in A \text{ and } q \in B, \text{ OR } p \in B \text{ and } q \in A, \\ 0, & \text{otherwise}, \end{cases}
\tag{50}
$$

for $p, q \in A \bigcup B$. Then, all entries of $\hat{M}$ are constants independent of $\gamma$. Let $\hat{\lambda}_1, \ldots, \hat{\lambda}_{2t}$ be the eigenvalues of $\hat{m}$. It follows that $\hat{\lambda}_1, \ldots, \hat{\lambda}_{2t}$ are also constants independent of $\gamma$.

Consider the matrix $M$ defined in equation 3. It follows from equation 47 and equation 49 that for any $p, q \in A \bigcup B$,

$$
M_{pq} = \begin{cases} \frac{z_1}{\|z\|}\left(\big(1 - \cos(\pi/t)\big) + \frac{\gamma}{z_1}2(\tilde{c} - 1)\right), & \text{if } p = q, \\ \frac{z_1}{\|z\|}\hat{m}_{p,q}, & \text{if } p \in A \text{ and } q \in B, \text{ OR } p \in B \text{ and } q \in A, \\ 0, & \text{otherwise}. \end{cases}
\tag{51}
$$

Consider the order $\big(a^0, \ldots, a^{t-1}, b^0, \ldots, b^{t-1}\big)$ on the elements of $A \bigcup B$. Then,

$$
M = \frac{z_1}{\|z\|}\left(\hat{M} + \frac{\gamma}{z_1}2(\tilde{c} - 1)I\right) = \frac{z_1}{\|z\|}\left(\hat{M} + \frac{2(\tilde{c} - 1)\cos(\pi/t)}{\tilde{c} - 1 + 1/\gamma}I\right),
\tag{52}
$$

where $I$ is the $2t \times 2t$ identity matrix and the second equality is from the definition of $z_1$ in equation 37. Denote the eigenvalues of $M$ by $\lambda_1, \ldots, \lambda_{2t}$. Then, from elementary linear algebra,

$$
\lambda_i = \frac{z_1}{\|z\|}\left(\hat{\lambda}_i + \frac{2(\tilde{c} - 1)\cos(\pi/t)}{\tilde{c} - 1 + 1/\gamma}\right)
\tag{53}
$$

for $i = 1, \ldots, 2t$. Therefore, there is at most one value of $\gamma$ for which $\lambda_i = 0$. Then, in view of equation 23, we have $\Pr(\lambda_i = 0) = 0$, over the random choice of $\gamma$. Thus, with probability one, $M$ has no zero eigenvalues and is thereby full-rank. As an immediate consequence, $M$ is full-rank for suitable choice of $\gamma$. This establishes Property (p8) and completes the proof of Proposition 1.

## D    PROOF OF LEMMA 2

Consider a block representation of $\theta$ as follows

$$\theta = \begin{bmatrix} \theta_1 \\ \vdots \\ \theta_m \end{bmatrix}, \tag{54}$$

where each $\theta_r$ is a $d$-dimensional vector.

It follows from the definition of $\delta$ that for any $t \in (0, \delta)$, if $w_r^T X_i > 0$, then $(w_r + \theta_r t)^T X_i > 0$; and if $w_r^T X_i < 0$, then $(w_r + \theta_r t)^T X_i < 0$. Therefore, for $r = 1, \ldots, m$ and $i = 1, \ldots, n$, and for any $t \in [0, \delta)$ ,

$$\mathbf{1}\big((w_r + \theta_r t)^T X_i \geq 0\big) = \mathbf{1}\big(w_r^T X_i > 0\big) + \mathbf{1}\big(w_r^T X_i = 0, \theta_r^T X_i \geq 0\big) \tag{55}$$

Consequently, for any $t \in [0, \delta]$ and $i = 1, \ldots, n$, we have

$$\begin{aligned} \hat{y}_i\big(w + \theta t, v\big) &= \sum_{r=1}^m v_r\big(w_r^T X_i + t\theta_r^T X_i\big)\mathbf{1}\big((w_r + \theta_r t)^T x_i \geq 0\big) \\ &= \sum_{r=1}^m v_r\big(w_r^T X_i + t\theta_r^T X_i\big)\Big(\mathbf{1}\big(w_r^T X_i > 0\big) + \mathbf{1}\big(w_r^T X_i = 0, \theta_r^T X_i \geq 0\big)\Big). \end{aligned} \tag{56}$$

It follows that $\hat{y}_i\big(w + \theta t, v\big)$ is a linear function of $t$ over the interval $t \in [0, \delta]$. Therefore, $F_\theta(t) = \sum_{i=1}^n \big(\hat{y}_i(w + \theta t, v) - y_i\big)^2$ is a quadratic and convex function of $t$, for $t \in [0, \delta]$.

## E    PROOF OF LEMMA 3

We first characterize active neurons for different inputs. For two subsets $S_1, S_2 \subset \mathbb{R}^d$ we let $S_1 \backslash S_2 = S_1 \bigcap S_2^c$. Recall the definition of $s_i(a)$ and $s_i(b)$ from Property (p6).

**Claim 1.** *For any $a \in A$, we have*

$$w_a^T X_{s_i(a)} = 0, \qquad i = 1, \ldots, d - 1, \tag{57}$$

$$w_a^T X_a > 0, \tag{58}$$

$$w_a^T X_b > 0, \qquad b \in \Big(B \bigcup \tilde{B}\Big) \backslash \{s_1(a), \ldots, s_{d-1}(a)\}, \tag{59}$$

$$w_a^T X_{B^+} > 0, \tag{60}$$

$$w_a^T X_{B^-} > 0, \tag{61}$$

$$w_a^T X_{A^+} < 0, \tag{62}$$

$$w_a^T X_{A^-} < 0, \tag{63}$$

$$w_a^T X_{a'} < 0, \qquad a' \in \Big(A \bigcup \tilde{A}\Big) \backslash \{a\}. \tag{64}$$

*Similarly, for any $b \in B$, we have*

$$w_b^T X_{s_i(b)} = 0, \qquad i = 1, \ldots, d - 1, \tag{65}$$

$$w_b^T X_b > 0, \tag{66}$$

$$w_b^T X_a > 0, \qquad a \in \Big(A \bigcup \tilde{A}\Big) \backslash \{s_1(b), \ldots, s_{d-1}(b)\}, \tag{67}$$

$$w_b^T X_{A^+} > 0, \tag{68}$$

$$w_b^T X_{A^-} > 0, \tag{69}$$

$$w_b^T X_{B^+} < 0, \tag{70}$$

$$w_b^T X_{B^-} < 0, \tag{71}$$

$$w_b^T X_{b'} < 0, \qquad b' \in \Big(B \bigcup \tilde{B}\Big) \backslash \{b\}. \tag{72}$$

*Proof of Claim 1.* Fix a $b \in B$. We begin by introducing some notations. Let $\mathcal{H}$ be the $(d-2)$-dimensional hyperplane in the $(d-1)$-dimensional space that passes through $s_1(b), \ldots, s_{d-1}(b)$. In the same spirit, let $\tilde{\mathcal{H}}$ be the $(d-1)$-dimensional subspace in the $d$-dimensional space that passes through $X_{s_1(b)}, \ldots, X_{s_{d-1}(b)}$, equivalently $\tilde{\mathcal{H}}$ is the subspace orthogonal to $w_b$. We denote the convex hull of $X_{s_1(b)}, \ldots, X_{s_{d-1}(b)}$ by $\mathcal{C}_B \triangleq \mathrm{Conv}\big(\{X_{s_1(b)}, \ldots, X_{s_{d-1}(b)}\}\big)$. Similarly, we let $\mathcal{Q}_A \triangleq \mathrm{conv}\big(\{X_a \mid a \in A \bigcup \tilde{A}\}\big)$ and $\mathcal{Q}_B \triangleq \mathrm{Conv}\big(\{X_b \mid b \in B \bigcup \tilde{B}\}\big)$.

Before presenting the proofs of properties equation 57–equation 72, we review make some easy observations. Recall the definition of $\epsilon_0$ from the paragraph proceeding equation 4. Let $\mathcal{B}_{\epsilon_0}^{d-1}$ be the intersection of the $\epsilon_0$-ball centered at 0 with the orthogonal space of $u_d$. Then, from the definition of $\epsilon$, we have

$$\mathcal{B}_{\epsilon_0}^{d-1} + u_d \subset \mathcal{Q}_A, \qquad \mathcal{B}_{\epsilon_0}^{d-1} - u_d \subset \mathcal{Q}_B. \tag{73}$$

For $x \in \mathbb{R}^d$, let $\pi(x)$ be the projection of $x$ on the span of first $d-1$ coordinates, i.e., the orthogonal space of $u_d$. Then,

$$
\begin{aligned}
\frac{\left\| \pi\left(X_{A^+}\right) \right\|}{\xi + 1/\beta - n - 1} &= \frac{\left\| \sum_{a \in A \bigcup \tilde{A}}(X_a - 1) \right\|}{\xi + 1/\beta - n - 1} \\
&= \frac{\left\| \sum_{a \in A \bigcup \tilde{A}} a \right\|}{\xi + 1/\beta - n - 1} \\
&< \frac{\left\| \sum_{a \in A \bigcup \tilde{A}} a \right\|}{\xi - n} \\
&< \frac{\left\| \sum_{a \in A \bigcup \tilde{A}} a \right\|}{\left\| \sum_{a \in A \bigcup \tilde{A}} a \right\|/\epsilon_0} \\
&= \epsilon_0,
\end{aligned}
\tag{74}
$$

where the first equality is from the definition of $X_{A^+}$ and the second inequality follows from the definition of $\xi$ in equation 4. Therefore, $\pi(X_{A^+})/(\xi + 1/\beta - n - 1) \in \mathcal{B}_{\epsilon_0}^{d-1}$. Consequently,

$$
\begin{aligned}
\frac{1}{\xi + 1/\beta - n - 1} X_{A^+} &= \frac{\pi(X_{A^+})}{\xi + 1/\beta - n - 1} + \frac{(u_d^T X_{A^+})u_d}{\xi + 1/\beta - n - 1} \\
&= \frac{\pi(X_{A^+})}{\xi + 1/\beta - n - 1} + \frac{(\xi + 1/\beta - n - 1)u_d}{\xi + 1/\beta - n - 1} \\
&= \frac{\pi(X_{A^+})}{\xi + 1/\beta - n - 1} + u_d \\
&\in \mathcal{B}_{\epsilon_0}^{d-1} + u_d \\
&\in \mathcal{Q}_A,
\end{aligned}
\tag{75}
$$

where the first equality is orthogonal decomposition of $X_{A^+}$, and last inclusion follows from equation 73. In the same vein, we can show that

$$-\frac{1}{\xi + n + 1 - 1/\beta} X_{B^+} \in \mathcal{Q}_A. \tag{76}$$

We proceed to verify equation 65–equation 72. Eq. equation 65 follows from equation 9. Recall the definitions $X_{A^-} = \xi u_d$ and $X_{B^-} = -\xi u_d$. Then, equation 10 implies equation 69 and equation 71.

Since $X_{s_1(b)}, \ldots, X_{s_{d-1}(b)}$ define a boundary of the $(d-1)$-dimensional convex set $\mathcal{Q}_A$, and $\tilde{\mathcal{H}}$ passes through $X_{s_1(b)}, \ldots, X_{s_{d-1}(b)}$, then all points in $\mathcal{Q}_A$ lie on a same side of $\tilde{\mathcal{H}}$. In other words, either we have $w_b^T x \geq 0$, for all $x \in \mathcal{Q}_A$; or we have $w_b^T x \leq 0$, for all $x \in \mathcal{Q}_A$. In view of Property (p3), $u_d \in \mathrm{int}(\mathcal{Q}_A)$. It then follows from equation 10 that for any $x \in \mathcal{Q}_A$, we have

$w_b^T x \geq 0$. Consequently, for any $x \in \mathcal{Q}_A \backslash \tilde{\mathcal{H}}$, we have $w_b^T x > 0$. In particular,

$$w_b^T X_a > 0, \qquad a \in \left( A \bigcup \tilde{A} \right) \backslash \{ s_1(b), \ldots, s_{d-1}(b) \},$$
$$w_b^T X_{A+} > 0,$$
$$-w_b^T X_{B+} > 0,$$

where the first inequality is because Property (p5) implies that $X_a \notin \tilde{\mathcal{H}}$ for $a \in \left( A \bigcup \tilde{A} \right) \backslash \{ s_1(b), \ldots, s_{d-1}(b) \}$, and the last two inequalities are due to equation 75 and equation 76, respectively. This establishes equation 67, equation 68, and equation 70.

For equation 66, it follows from Property (p4) that $b$ and the origin, $0$, lie on opposite sides of hyperplane $\mathcal{H}$. Consequently, $-X_b$ and $u_d$ also lie on opposite sides of hyperplane $\tilde{\mathcal{H}}$. Therefore, $w_b^T X_b$ and $w_b^T u_d$ have a same sing. It then follows from equation 10 that $w_b^T X_b > 0$, and equation 66 follows.

For equation 72, it follows from Property (p7) that for any $b' \in B \bigcup \tilde{B}$ with $b' \neq b$, $X_b$ and $X_{b'}$ lie on opposite sides of $\tilde{\mathcal{H}}$. Eq. equation 66 then implies that $w_b^T X_{b'} < 0$. This establishes equation 72, and completes the proof of Claim 1. $\qquad \square$

In light of Claim 1, it is easy to see for $r = 1, \ldots, m$ and $i = 1, \ldots, n$ that if $w_r^T X_i = 0$, then

$$v_r e_i = \frac{1}{\sqrt{m}}. \tag{77}$$

In our next claim, we examine a linear combination of data points for which a particular neuron is active.

**Claim 2.** *For $r = 1, \ldots, m$, there exist constants $\gamma_1^r, \ldots, \gamma_m^r$ such that*

$$\sum_{i=1}^n \gamma_i^r e_i X_i \mathbf{1}\left( w_r^T X_i = 0 \right) + \sum_{i=1}^n e_i X_i \mathbf{1}\left( w_r^T X_i > 0 \right) = 0. \tag{78}$$

*Proof of Claim 2.* Fix a $b \in B$. We prove the claim for the neuron associated to $b$. It follows from Claim 1 that

$$\sum_{i=1}^n e_i X_i \mathbf{1}\left( w_b^T X_i > 0 \right) = e_b X_b + \sum_{a \in (A \bigcup \tilde{A}) \backslash \left\{ s_1(b), \ldots, s_{d-1}(b) \right\}} e_a X_a + e_{A+} X_{A+} + e_{A-} X_{A-}$$

$$= -X_b + \sum_{a \in (A \bigcup \tilde{A}) \backslash \left\{ s_1(b), \ldots, s_{d-1}(b) \right\}} X_a + \left( X_{A+} - X_{A-} \right)$$

$$= -X_b + \sum_{a \in (A \bigcup \tilde{A}) \backslash \left\{ s_1(b), \ldots, s_{d-1}(b) \right\}} X_a - \left( \sum_{a \in A \bigcup \tilde{A}} X_a - (1/\beta - 1) u_d \right)$$

$$= -X_b - \sum_{i=1}^{d-1} X_{s_i(b)} + (1/\beta - 1) u_d, \tag{79}$$

where the second equality is due to the definitions of $e_a$, $e_b$, $e_{A+}$, and $e_{A-}$, and the third equality is from the definitions of $X_{A+}$ in equation 5.

On the other hand, it follows from Property (p6) that there exist scalars $\alpha_1, \ldots, \alpha_d \in (0, 1)$ such that $b = \sum_{i=1}^{d-1} \alpha_i s_i(b)$. Therefore, from the definition of $X_b$,

$$-\left( X_b + u_d \right) = \sum_{i=1}^{d-1} \alpha_i \left( X_{s_i(b)} - u_d \right) \tag{80}$$

Moreover, Property (p4) implies that $\sum_{i=1}^{d-1} \alpha_i = 1/\beta$. Then, from equation 80,

$$-X_b = \sum_{i=1}^{d-1} \alpha_i X_{s_i(b)} - \left(\sum_{i=1}^{d-1} \alpha_i - 1\right) u_d = \sum_{i=1}^{d-1} \alpha_i X_{s_i(b)} - (1/\beta - 1) u_d. \tag{81}$$

For $i = 1, \ldots, d-1$, let $\gamma_i = 1 - \alpha_i$. Then, Claim 1 implies that

$$\begin{aligned}
\sum_{i=1}^{n} \gamma_i e_i X_i \mathbf{1}\left(w_b^T X_i = 0\right) &= \sum_{i=1}^{d-1} \gamma_i e_i X_{s_i(b)} \\
&= \sum_{i=1}^{d-1} (1 - \alpha_i) X_{s_i(b)} \\
&= \sum_{i=1}^{d-1} X_{s_i(b)} - \sum_{i=1}^{d-1} \alpha_i X_{s_i(b)} \\
&= \sum_{i=1}^{d-1} X_{s_i(b)} + X_b - \left(\frac{1}{\beta} - 1\right) u_d,
\end{aligned} \tag{82}$$

where last equality in due to equation 81. Combing equation 79 and equation 82, we obtain equation 78 for $w_r = w_b$. A similar argument implies equation 78 for $w_r = w_a$, $a \in A$. This completes the proof of Claim 2. □

Back to the proof of Lemma 3, for $r = 1, \ldots, m$, let

$$\epsilon_r \triangleq \min_{i=1,\ldots,d-1} \left( \min\left(\gamma_i^r, (1 - \gamma_i^r)\right)\right), \tag{83}$$

for the constant $\gamma_i^r$ defined in Claim 2. It follows that $\epsilon_r > 0$, for $r = 1, \ldots, m$.

For any $r \le m$, $X_{s_1(r)}, \ldots, X_{s_{d-1}(r)}$ are linearly independent and, by definition, are all orthogonal to $w_r$. Therefore, there exists a constant $\epsilon_r' > 0$ such that for any $\zeta_r \in \mathbb{R}^d$ with $\|\zeta_r\| = 1$, we have $\max_{i=1,\ldots,d-1} |\zeta_r^T X_{s_i(r)}| \ge \epsilon_r' \|\zeta_r^\perp\|$, where $\zeta_r^\perp$ is the projection of $\zeta_r$ on the null-space of $w_r$. Consequently, for any $\zeta_r \in \mathbb{R}^d$ with $\|\zeta_r\| = 1$,

$$\max_{i=1,\ldots,n} |\zeta_r^T X_i| \times \mathbf{1}\left(w_r^T X_i = 0\right) \ge \epsilon_r' \|\zeta_r^\perp\|.$$

In particular, considering the block-vector representation of $\theta$ in equation 54, we obtain for $r = 1, \ldots, m$,

$$\max_{i=1,\ldots,n} |\theta_r^T X_i| \times \mathbf{1}\left(w_r^T X_i = 0\right) \ge \max_{i=1,\ldots,n} \epsilon_r' \|\theta_r^\perp\|. \tag{84}$$

Let $\mu \triangleq \min_r \epsilon_r \epsilon'_r / \sqrt{m}$. Then, $\mu > 0$. It then follows from Claim 2 that, for $r = 1, \ldots, m$,

$$
\begin{aligned}
v_r \sum_{i=1}^{n} e_i \theta^T X_i \mathbf{1}\big(w_r^T X_i = 0, \theta_r^T X_i \geq 0\big) &+ v_r \sum_{i=1}^{n} e_i \theta_r^T X_i \mathbf{1}\big(w_r^T X_i > 0\big) \\
&= v_r \sum_{i=1}^{n} e_i \theta_r^T X_i \mathbf{1}\big(w_r^T X_i = 0, \theta_r^T X_i \geq 0\big) - v_r \sum_{i=1}^{n} \gamma_i^r e_i \theta_r^T X_i \mathbf{1}\big(w_r^T X_i = 0\big) \\
&= v_r \sum_{i=1}^{n} e_i \theta_r^T X_i \Big(\mathbf{1}(\theta_r^T X_i \geq 0) - \gamma_i^r\Big) \mathbf{1}\big(w_r^T X_i = 0\big) \\
&= \frac{1}{\sqrt{m}} \sum_{i=1}^{n} \theta_r^T X_i \Big(\mathbf{1}(\theta_r^T X_i \geq 0) - \gamma_i^r\Big) \mathbf{1}\big(w_r^T X_i = 0\big) \\
&= \frac{1}{\sqrt{m}} \sum_{i=1}^{n} \big|\theta_r^T X_i\big| \times \big|\mathbf{1}(\theta_r^T X_i > 0) - \gamma_i^r\big| \times \mathbf{1}\big(w_r^T X_i = 0\big) \\
&\geq \frac{1}{\sqrt{m}} \epsilon_r \sum_{i=1}^{n} \big|\theta_r^T X_i\big| \mathbf{1}\big(w_r^T X_i = 0\big) \\
&\geq \frac{\epsilon_r}{\sqrt{m}} \max_{i=1,\ldots,n} \big|\theta_r^T X_i\big| \mathbf{1}\big(w_r^T X_i = 0\big) \\
&\geq \frac{\epsilon_r}{\sqrt{m}} \epsilon'_r \|\theta_r^\perp\| \\
&\geq \mu \|\theta_r^\perp\|,
\end{aligned}
\tag{85}
$$

where the third equality is due to equation 77, the fourth equality is because $\theta_r^T X_i$ and $\mathbf{1}(\theta_r^T X_i > 0) - \gamma_i^r$ have always the same sign, the first inequality is by definition of $\epsilon_r$ in equation 83, the third inequality follows from equation 84, and the last inequality is from the definition of $\mu$.

On the other hand, equation 2 implies that

$$
\begin{aligned}
\frac{dF_\theta(t)}{d^+t}\bigg|_{t=0} &= \lim_{t \downarrow 0} \sum_{r=1}^{m} v_r \theta_r^T \sum_{i=1}^{n} e_i X_i \mathbf{1}\big((w_r + \theta_r t)^T X_i \geq 0\big) \\
&= \sum_{r=1}^{m} v_r \sum_{i=1}^{n} \theta_r^T e_i X_i \Big(\mathbf{1}\big(w_r^T X_i = 0, \theta_r^T X_i \geq 0\big) + \mathbf{1}\big(w_r^T X_i > 0\big)\Big) \\
&= \sum_{r=1}^{m} \left(v_r \sum_{i=1}^{n} e_i \theta^T X_i \mathbf{1}\big(w_r^T X_i = 0, \theta_r^T X_i \geq 0\big) + v_r \sum_{i=1}^{n} e_i \theta_r^T X_i \mathbf{1}\big(w_r^T X_i > 0\big)\right) \\
&\geq \sum_{r=1}^{m} \mu \|\theta_r^\perp\| \\
&\geq \mu \|\theta^\perp\|
\end{aligned}
\tag{86}
$$

where the second equality is due to equation 55 and the first inequality follows from equation 85. This completes the proof of Lemma 3.

## F  PROOF OF LEMMA 4

We begin by a claim. Given a $q \in A \bigcup B$, recall the definition of hyperplane $\mathcal{H}_q$ from Property (p7).

**Claim 3.** *For any pair of points $p, q \in A \bigcup B$, we have*

$$
\big|w_q^T X_p\big| = \frac{1}{\sqrt{1 + d(0, \mathcal{H}_q)^2}} \, d\big(p, \mathcal{H}_q\big).
\tag{87}
$$

*Proof of Claim 3.* In the $(d-1)$-dimensional space, let $\omega$ be the unit normal vector of $\mathcal{H}_q$. Recall from Property (p6) that $s_1(q), \ldots, s_{d-1}(q)$ are located on $\mathcal{H}_q$. Let,

$$\gamma \triangleq \omega^T s_1(q) = d(0, \mathcal{H}_q). \tag{88}$$

Then,

$$d(p, \mathcal{H}_q) = |\omega^T p - \gamma|. \tag{89}$$

Without loss of generality suppose that $q \in A$. Let $\tilde{\omega}$ be the lifting of $\omega$ from the $(d-1)$-dimensional space to the $d$-dimensional space by appending $\omega$ by a new coordinate with zero coefficient, i.e., $\tilde{\omega}$ is a $d$-dimensional vector with $\tilde{\omega}_i = \omega_i$, for $i = 1, \ldots, d-1$, and $\tilde{\omega}_d = 0$. For $\gamma$ defined in equation 88, let

$$z \triangleq \frac{1}{\sqrt{1+\gamma^2}} (\tilde{\omega} - \gamma u_d). \tag{90}$$

Then, we have $\|z\| = 1$ and $z^T u_d < 0$. Moreover, for $i = 1, \ldots, d-1$,

$$\begin{aligned}
z^T X_{s_i(q)} &= \frac{1}{\sqrt{1+\gamma^2}} \left( \tilde{\omega}^T X_{s_i(q)} - \gamma u_d^T X_{s_i(q)} \right) \\
&= \frac{1}{\sqrt{1+\gamma^2}} \left( -\omega^T s_i(q) + \gamma u_d^T u_d \right) \\
&= \frac{1}{\sqrt{1+\gamma^2}} (-\gamma + \gamma) \\
&= 0,
\end{aligned} \tag{91}$$

where the second equality is because $s_i(q) \in B \bigcup \tilde{B}$ for $q \in A$. It follows from equation 91 and the definition of $w_q$ in equation 7 and equation 8 that $w_q = z$. Therefore,

$$\begin{aligned}
\left| w_q^T X_p \right| &= \left| z^T X_p \right| \\
&= \frac{1}{\sqrt{1+\gamma^2}} \left| \tilde{\omega}^T X_p - \gamma u_d^T X_p \right| \\
&= \frac{1}{\sqrt{1+\gamma^2}} \left| \omega^T p - \gamma u_d^T u_d \right| \\
&= \frac{1}{\sqrt{1+\gamma^2}} \left| \omega^T p - \gamma \right| \\
&= \frac{1}{\sqrt{1+d(0, \mathcal{H}_q)^2}} d(p, \mathcal{H}_q),
\end{aligned} \tag{92}$$

where the second equality is from the definition of $z$ in equation 90, and the last equality is due to equation 88 and equation 89. This completes the proof of Claim 3. $\square$

We now proceed to the proof of Lemma 4. Fix an arbitrary $\theta \in \mathbb{R}^d$ with $\|\theta\| = 1$. Without loss of generality[4] assume that $F(\cdot, \cdot)$ is differentiable at $(w + \delta\theta/2, v)$. For $r = 1, \ldots, m$, let $J_r$ be a diagonal matrix whose $(i, i)$ entry, for $i = 1, \ldots, n$, equals

$$\mathbf{1}\left( (w_r + \theta_r t)^T X_i \geq 0 \right) = \mathbf{1}\left( w_r^T X_i > 0 \right) + \mathbf{1}\left( w_r^T X_i = 0, \theta_r^T X_i \geq 0 \right), \tag{93}$$

where the equality is due to equation 55. Recall the definition of matrix $G$ in equation 18:

$$G = \left[ \ v_1 J_1 \mathbf{X}^T \ | \ \cdots \ | \ v_m J_m \mathbf{X}^T \ \right]. \tag{94}$$

Then, for any $t \in (0, \delta)$,

$$\frac{d^2 F_\theta(t)}{dt^2} = \frac{d^2 F(w + t\theta, v)}{dt^2} = \theta^T \nabla^2_{w+t\theta} F(w + t\theta, v) \theta = \theta^T G^T G \theta = \|G\theta\|^2, \tag{95}$$

---

[4]This is because otherwise, $(w + \delta\theta/2, v)$ is the limit point of a sequence of weights at which $F$ is differentiable. Since our arguments carry over to the weights in this sequence, we can conclude that all bounds also apply to their limit point, $(w + \delta\theta/2, v)$.

where the third equality is due to equation 19. On the other hand, since $\theta = \theta^\| + \theta^\perp$, we have

$$G\theta = G\theta^\| + G\theta^\perp. \tag{96}$$

In the following claim, we elaborate on $\|G\theta^\|\|$.

**Claim 4.** *There exists a constant $\eta_1 > 0$ such that $\|G\theta^\|\|^2 \geq 2\eta_1\|\theta^\|\|^2$, for all $\theta \in \mathbb{R}^d$.*

*Proof of Claim 4.* Recall that $\theta^\|$ is the projection of $\theta$ on subspace $H_w$ defined in equation 14. Then, there exist constants $\alpha_1 \ldots, \alpha_m$ such that

$$\theta^\| = \begin{bmatrix} \alpha_1 w_1 \\ \vdots \\ \alpha_m w_m \end{bmatrix}. \tag{97}$$

Let $\alpha$ be the vector representation of $\alpha_1, \ldots, \alpha_m$. Then,

$$\|\theta^\|\|^2 = \sum_{i=1}^m \alpha_i^2 \|w_i\|^2 = \sum_{i=1}^m \alpha_i^2 = \|\alpha\|^2. \tag{98}$$

Consider the $n \times m$ matrix

$$\tilde{G} = \begin{bmatrix} v_1 J_1 \mathbf{X}^T w_1 & \cdots & v_m J_m \mathbf{X}^T w_m \end{bmatrix}. \tag{99}$$

Then, from the definition of matrix $G$ in equation 18,

$$\begin{aligned} G\theta^\| &= \begin{bmatrix} v_1 J_1 \mathbf{X}^T & \cdots & v_m J_m \mathbf{X}^T \end{bmatrix} \begin{bmatrix} \alpha_1 w_1 \\ \vdots \\ \alpha_m w_m \end{bmatrix} \\ &= \begin{bmatrix} v_1 J_1 \mathbf{X}^T w_1 & \cdots & v_m J_m \mathbf{X}^T w_m \end{bmatrix} \alpha \\ &= \tilde{G}\alpha. \end{aligned} \tag{100}$$

Each column of $\tilde{G}$ corresponds to a neuron, and thereby is associated to a point in $A \bigcup B$. In the same vein, every row of $\tilde{G}$ is associated to an input vector. By removing some rows of $\tilde{G}$, we devise a matrix $\tilde{M}$ so that each row of $\tilde{M}$ is associated to an input $X_p$ for $p \in A \bigcup B$. Therefore, $\tilde{M}$ is an $m \times m$ matrix, whose rows and columns are associated to the points in $A \bigcup B$. It follows that $\tilde{M}\alpha$ is a vector obtained by removing some entries from vector $\tilde{G}\alpha$. As a result,

$$\|\tilde{M}\alpha\| \leq \|\tilde{G}\alpha\|. \tag{101}$$

In the following, we capitalize on Property (p8) to show that $\tilde{M}$ is full-rank.

For $q \in A \bigcup B$, let

$$\gamma_q \triangleq \frac{v_q}{\sqrt{1 + d(0, \mathcal{H}_q)}}.$$

For $p, q \in A \bigcup B$, the entry in row $p$ and column $q$ of $\tilde{M}$ equals

$$\begin{aligned} \tilde{M}_{pq} &= \tilde{G}_{pq} \\ &= v_q X_p^T w_q \left( \mathbf{1}(w_q^T X_p > 0) + \mathbf{1}(w_q^T X_p = 0, \theta_q^T X_p \geq 0) \right) \\ &= v_q X_p^T w_q \mathbf{1}(w_q^T X_p > 0) \\ &= \frac{v_q}{\sqrt{1 + d(0, \mathcal{H}_q)}} d(p, \mathcal{H}_q) \mathbf{1}(w_q^T X_p > 0) \\ &= \gamma_q d(p, \mathcal{H}_q) \mathbf{1}(w_q^T X_p > 0). \end{aligned} \tag{102}$$

where the first equality is from the definition of $\tilde{M}$, the second equality follows from the definitions of $\tilde{G}$ and $J_q$ in equation 99 and equation 93, the third equality is because $X_p^T w_q \mathbf{1}(w_q^T X_p = 0) = 0$, and the fourth equality is due to Claim 3. Then, Claim 1 implies that for any $p, q \in A \bigcup B$,

$$\tilde{M}_{pq} = \begin{cases} \gamma_q d(p, \mathcal{H}_q), & \text{if } p = q, \text{ OR } p \in A \text{ and } q \in B, \text{ OR } p \in B \text{ and } q \in A, \\ 0, & \text{otherwise.} \end{cases} \tag{103}$$

Compared to matrix $M$ defined in equation 3, each column $q$ of $\tilde{M}$ equals the column $q$ of $M$ multiplied by a non-zero constant $\gamma_q$. In view of Property (p8), $M$ is full-rank. It follows that $\tilde{M}$ is full-rank, as well.

Let $\sigma$ be the smallest singular value of $\tilde{M}$. Since $\tilde{M}$ is full-rank, we have $\sigma > 0$. Moreover,

$$\|\tilde{M}\alpha\| \geq \sigma\|\alpha\|. \tag{104}$$

Then,

$$\|G\theta^\|\|^2 = \|\tilde{G}\alpha\|^2 \geq \|\tilde{M}\alpha\|^2 \geq \sigma^2\|\alpha\|^2 = \sigma^2\|\theta^\|\|^2, \tag{105}$$

where the equations are due to equation 100, equation 101, equation 104, and equation 98, respectively. Claim 4 then follows for $\eta_1 = \sigma^2/2$. $\square$

Back to the proof of Lemma 4, we denote by $\sigma_{\max}$ the largest singular value of $G$. Let $\eta_2 \triangleq \sigma_{\max}^2$. Then,

$$
\begin{aligned}
\|G\theta\|^2 &= \left\|G\theta^\| + G\theta^\perp\right\|^2 \\
&= \left\|G\theta^\|\right\|^2 + \left\|G\theta^\perp\right\|^2 + 2\left(G\theta^\|\right)^T\left(G\theta^\perp\right) \\
&\geq \left\|G\theta^\|\right\|^2 - 2\left\|G\theta^\|\right\| \times \left\|G\theta^\perp\right\| \\
&\geq \left\|G\theta^\|\right\|^2 - 2\sigma_{\max}\|\theta^\|\| \times \sigma_{\max}\|\theta^\perp\| \\
&\geq \left\|G\theta^\|\right\|^2 - 2\sigma_{\max}^2\|\theta^\perp\| \\
&= \left\|G\theta^\|\right\|^2 - 2\eta_2\|\theta^\perp\| \\
&\geq 2\eta_1\left\|\theta^\|\right\|^2 - 2\eta_2\|\theta^\perp\|,
\end{aligned}
\tag{106}
$$

where the second inequality is from the definition of $\sigma_{\max}$, the third inequality is because $\|\theta^\|\| \leq \|\theta\| = 1$, the last equality is by the definition of $\eta_2$, and the last inequality follows from Claim 4. This completes the proof of Lemma 4.

## G  PROOF OF LEMMA 5

Fix a $\theta \in \mathbb{R}^{md}$ with $\|\theta\| = 1$. Recall the definition of $x_0$ from the paragraph proceeding equation 17. If $\|\theta^\perp\| \geq x_0$, then for any $t \in [0, \delta]$,

$$
\begin{aligned}
F_\theta(t) - F_\theta(0) &\geq F_\theta'(0)t \\
&\geq \mu\|\theta^\perp\|t \\
&\geq \mu x_0 t \\
&= \epsilon\delta t \\
&\geq \epsilon t^2,
\end{aligned}
$$

where the first inequality is from convexity of $F_\theta$ in Lemma 2, the second inequality is due to Lemma 3, the equality is by the definition of $\epsilon$ in equation 17, and the last inequality is because $t \leq \delta$.

On the other hand, if $\|\theta^\perp\| < x_0$, then for any $t \in [0, \delta]$,

$$
\begin{aligned}
F_\theta(t) - F_\theta(0) &= F_\theta'(0)t + \frac{1}{2}F_\theta''t^2 \\
&\geq \frac{1}{2}F_\theta''t^2 \\
&\geq \left(\eta_1\|\theta^\|\|^2 - \eta_2\|\theta^\perp\|\right)t^2 \\
&= \left(\eta_1\left(1 - \|\theta^\perp\|^2\right) - \eta_2\|\theta^\perp\|\right)t^2 \\
&< \left(\eta_1(1 - x_0^2) - \eta_2 x_0\right)t^2 \\
&= \frac{\mu x_0}{\delta}t^2 \\
&= \epsilon t^2,
\end{aligned}
$$

where the first equality is because $F_\theta$ is quadratic (c.f. Lemma 2), the first inequality follows from Lemma 3, the second inequality is due to Lemma 4, the third inequality is because $\|\theta^\perp\| < x_0$, and the last two equalities are due to $p(x_0) = 0$ and the definition of $\epsilon$ in equation 17, respectively. Combining the above two cases, we obtain Lemma 5.

