# OpenReview forum: "Bounds on Over-Parameterization for Guaranteed Existence of Descent Paths in Shallow ReLU Networks"
_ICLR.cc/2020/Conference — Accept (Poster)_

### Official Review · AnonReviewer1 · 2019-10-23
**Official Blind Review #1**

**Rating:** 6

**Review:**

This paper analyzes the existence of descent paths from any initial point to the global minimum for the two-layer ReLU network and gives a better characterization of the network width that guarantees the descent path property. Concretely, the paper shows that there exists poor local minima under the case of $n > m+2d-2$ by constructing concrete examples of datasets.

To show the global convergence property of the optimization method, this kind of landscape analysis is very important. Basically, I like this paper and I think it makes a certain contribution to this line of researches. However, I did not verify the proof.

A few questions:
- I am not sure why the authors say that "it was not known whether the descent path property holds for $m \in (2n/d, n)$" by citing [Soudry and Hoffer(2017)]. I think [Soudry and Hoffer(2017)] does not mention the descent path property. Is this my misunderstanding?
- The theory is limited to 2-layer ReLU. Can it be extended to deep networks?
- The datasets producing poor local minima seem quite artificial. Does this theory hold for a more natural setting (e.g., assume a true distribution or function having preferable properties)?

Typos:
- In abstruct: exit -> exist
- Section 1.2: $m < n - 2d$ -> $m < n - 2d + 2$.
- After Corollary 1: Note that 1 does not ... -> Note that Corollary 1 does not

-----
Update:
I thank the authors for the response. My concerns have been well addressed and my review stands. I would like to keep the score.

**Experience Assessment:**

I have read many papers in this area.

**Review Assessment: Checking Correctness Of Derivations And Theory:**

I assessed the sensibility of the derivations and theory.

**Review Assessment: Checking Correctness Of Experiments:**

N/A

**Review Assessment: Thoroughness In Paper Reading:**

N/A

---

> ### Author Response · Authors · 2019-11-12
> **Author response to Review #1**
>
> Thank you for meticulous reading and your overall positive feedback. Please see below for our responses to the questions.
>
> - Regarding the reference [Soudry and Hoffer(2017)]
>
> Yes, [Soudry and Hoffer(2017)] does not mention the descent path property. We will revise the statement in the following form: “Prior to the present work, it was not known whether the “descent path property” holds for $m<n$. Even for $m\in (2n/d,n)$, where zero empirical risk is known to be achievable (Soudry and Hoffer, 2017), the existence of descent paths was in question.”
>
> - Regarding the extension of our results to deep networks
>
> This is an interesting question, to which we do not have a definite answer. Actually, as network depth increases, the loss surface spontaneously transitions from fairly benign to chaotic (Li, et al. 2018). On the plus side, we already know that a $k$-layer network of width $m$ has descent path property if $n\le m$ (Yun, et al. 2018). This is because in this case, the weights of the last layer suffice to form a descent path to zero loss. To the best of our knowledge, $n\le m$ is the best known bound for guaranteed existence of descent paths in deep networks. However, chances are that the property carries over to much smaller network widths. This we do not know, and needs to be addressed in future works.
>
> - Regarding applicability of
>
> This is a very important open problem, as discussed in the Discussion section. We have strong indications that the statement of Corollary 1 holds with high probability over random data sets (the probability tends to 1 as m=n-1 grows large). This claim, on which we are currently working, relies heavily on the intuitions and tools we developed in this work. More specifically, the argument in Appendix A shows that every cupped min is the intersection of $(m-1)d$ differentiable surface. For random datasets, several such intersections emerge in the loss landscape, and with high probability some of them have the shape of a cupped min (i.e. the landscape is locally curved upwards). This we will elaborate on more in an upcoming publication.
>
>
> - Regarding typos
>
> Thanks very much for reading our work meticulously. We correct all typos in the revised version.
>
> Li, Hao, et al. "Visualizing the loss landscape of neural nets." Advances in Neural Information Processing Systems. 2018.
> Yun, Chulhee, et al. "Small relu networks are powerful memorizers: a tight analysis of memorization capacity." arXiv preprint, arXiv: 1810.07770. 2018.

---

### Official Review · AnonReviewer2 · 2019-10-23
**Official Blind Review #2**

**Rating:** 6

**Review:**

This paper studies the landscape properties of over-parameterized two-layer neural networks, and proposes a network width lower bound for guaranteed existence of descent paths that is tighter than existing results. In particular, the authors prove that if the network width m \leq n - 2d, then there exist training data sets and initial weights such that the square loss on the neural network has no descent path connecting the initial weights and global minima.

Overall, I think this paper is of good quality. The presentation is clear and the logic is easy to follow. I did some high-level check and the theoretical analysis seems reasonable. However, I have the following questions:

1. As is stated in Corollary 1 and discussed below Theorem 1, the training sample sets that lead to suboptimal capped minima are not of measure-zero. However, it seems that no rigorous proof is provided for this result. Perhaps I have missed something, but is Corollary 1 straight forward given Theorem 1 and Lemma 1? How large is the probability? It would be better if the authors could provide a more rigorous proof.

2. A recent line of work has shown convergence of gradient-based algorithms for over-parameterized neural networks. It would be interesting if the authors could provide more comparison between this paper and the results of these works:

- Simon S. Du, Jason D. Lee, Haochuan Li, Liwei Wang, Xiyu Zhai, Gradient Descent Finds Global Minima of Deep Neural Networks
- Zeyuan Allen-Zhu, Yuanzhi Li, Zhao Song, A Convergence Theory for Deep Learning via Over-Parameterization
- Difan Zou, Yuan Cao, Dongruo Zhou, Quanquan Gu, Stochastic Gradient Descent Optimizes Over-parameterized Deep ReLU Networks


**Experience Assessment:**

I have published one or two papers in this area.

**Review Assessment: Checking Correctness Of Derivations And Theory:**

I assessed the sensibility of the derivations and theory.

**Review Assessment: Checking Correctness Of Experiments:**

N/A

**Review Assessment: Thoroughness In Paper Reading:**

I read the paper at least twice and used my best judgement in assessing the paper.

---

> ### Author Response · Authors · 2019-11-12
> **Author response to Review #2**
>
> We thank you for your careful review and insightful feedback. Please see below for our responses to the questions.
>
> 1- Regarding Corollary 1
>
> The proof in the paper is pretty fast. The corollary is not straight forward. Below is a more detailed proof sketch. Consider a dataset S and denote its corresponding landscape by $F_S$. Note that a continuous change in $S$ results in a continuous change in $F_S$. From Theorem 1, let $w_0$ be a suboptimal cupped min of $F_S$. Then, from the definition of cupped min, there is a $\delta>0$, such that for any weight $w$ with $\|w-w_0\|=\delta$, we have $F_S(w)>F_S(w_0)+\epsilon$. In other words, any path that starts at $w_0$ and end up in the set of global minima of $F_S$ would have an uphill climb of at least $\epsilon$. Therefore, it follows from the continuity of $F_S$ with respect to $S$ that under a small enough perturbation of the dataset, there would still be an ($\epsilon/2$)-uphill-climb in all paths that start at $w_0$ and end up in a global min. Equivalently, the descent path property remains violated for a small neighborhood of S with non-zero measure.
>
> Following the above idea, we actually have a rigorous proof that involves tedious mathematical analysis arguments. As the corollary is not the main point of the paper and the intuition provided in the paper is hopefully pretty convincing, we decided not to include a rigorous proof. However, if the expert reviewer suggests that it will improve the clarity of the paper, we would be happy to include the rigorous proof in the appendix. In any case, we discuss the proof sketch in more details to clarify the idea in the revised version.
>
> As for the probability that the descent path property is violated, Corollary 1 shows that the probability is non-zero, and provides no further information about the size of the probability. The exact probability of existence of sub-optimal cupped minima is an interesting non-trivial problem. The results of this paper fall short of resolving this problem, however the methodology and ideas in the proofs provide promising grounds to attack the problem. This constitutes a current direction of our research, and we are working on a promising idea to show that for $m=n-1$ as $m$ grows large the probability of existence of suboptimal cupped minima tends to one.
>
>
> 2- Regarding to the connection of our work with the literature on convergence of GD
>
> This paper was actually inspired by the line of work mentioned by the respected reviewer. The best known guarantees for convergence of gradient descent methods in shallow networks require extremely large widths $m>n^8$. Based on our experience with that problem, we think that $m>n/\gamma$ (for some constant $\gamma$ that depends on pairwise distances of data points) should suffice for a theoretical guarantee of convergence of GD.
>
> On the other hand, empirical results show that gradient descent typically achieve zero-loss for networks of widths as small as $m=n/d$ [Oymak and Soltanolkotabi, 2019]. It is also theoretically guaranteed that in this regime there exist parameters that achieve zero loss [Soudry and Hoffer, 2017].
>
> The above literature is about trying to find the smallest over-parameterization under which optimization is easy. This gives rise to the following question. What exactly is an over-parameterized network? In other words, which regimes among $m=n$, $m=n/d$, and $m=n^8$ should be considered as over-parameterized? This paper addresses this problem from the viewpoint of descent algorithms. We show that the edge of over-parameterization (in the sense of descent path property) lies sharp at $m=n$.
>
> As for convergence of GD, existence of descent paths from a given starting point is a necessary condition. However, the mere non-existence of descent path is far from sufficient for guaranteed polynomial time convergence of GD. The current paper neither aims to nor does solve the problem of convergence of gradient descent methods. However, we believe that a good understanding of shape and properties of loss landscape will eventually prove useful in performance guarantees of GD. The current work tries to take a step in that direction.
>
> Oymak, Samet, and Mahdi Soltanolkotabi. "Towards moderate overparameterization: global convergence guarantees for training shallow neural networks." arXiv preprint arXiv:1902.04674, 2019.
> Daniel Soudry and Elad Hoffer. Exponentially vanishing sub-optimal local minima in multilayer
> neural networks. arXiv preprint arXiv:1702.05777, 2017.

---

### Decision · Program_Chairs · 2019-12-19

**Decision:**

Accept (Poster)

**Comment:**

This article investigates the optimization landscape of shallow ReLU networks, showing that for sufficiently narrow networks there are data sets for which there is no descent paths to the global minimiser. The topic and the nature of the results is very interesting. The reviewers found that this article makes important contributions in a relevant line of investigation and had generally positive ratings. The authors' responses addressed questions from the initial reviews, and the discussion helped identifying questions for future study departing from the present contribution.